# Complementary encoding of spatial information in hippocampal astrocytes

Sebastiano Curreli[1,2], Jacopo Bonato[2,3,4], Sara Romanzi[1,2,5], Stefano Panzeri[2,3,6]*, Tommaso Fellin[1,2]*

1 Optical Approaches to Brain Function Laboratory, Istituto Italiano di Tecnologia, Genova, Italy, 2 Neural Coding Laboratory, Istituto Italiano di Tecnologia, Genova, Italy, 3 Neural Computation Laboratory, Istituto Italiano di Tecnologia, Rovereto, Italy, 4 Department of Pharmacy and Biotechnology, University of Bologna, Bologna, Italy, 5 University of Genova, Genova, Italy, 6 Department of Excellence for Neural Information Processing, Center for Molecular Neurobiology (ZMNH), University Medical Center Hamburg-Eppendorf (UKE), Hamburg, Germany

* s.panzeri@uke.de (SP); tommaso.fellin@iit.it (TF)

**Data Availability Statement:** All relevant data are within the paper and its Supporting Information files.

## Abstract

Calcium dynamics into astrocytes influence the activity of nearby neuronal structures. However, because previous reports show that astrocytic calcium signals largely mirror neighboring neuronal activity, current information coding models neglect astrocytes. Using simultaneous two-photon calcium imaging of astrocytes and neurons in the hippocampus of mice navigating a virtual environment, we demonstrate that astrocytic calcium signals encode (i.e., statistically reflect) spatial information that could not be explained by visual cue information. Calcium events carrying spatial information occurred in topographically organized astrocytic subregions. Importantly, astrocytes encoded spatial information that was complementary and synergistic to that carried by neurons, improving spatial position decoding when astrocytic signals were considered alongside neuronal ones. These results suggest that the complementary place dependence of localized astrocytic calcium signals may regulate clusters of nearby synapses, enabling dynamic, context-dependent variations in population coding within brain circuits.

## Introduction

Astrocytes, the most abundant class of glial cells in the brain, exhibit complex dynamics in intracellular calcium concentration [1]. Intracellular calcium signals can be spatially restricted to individual subcellular domains (e.g., cellular processes versus somata) and be coordinated across astrocytic cells [2–8]. In the intact brain, astrocytic calcium dynamics can be spontaneous [9] or triggered by the presentation of external physical stimuli [4,7,10–12]. Interestingly, previous reports suggest that astrocytic calcium signals triggered by external sensory stimuli largely mirror the activity of local neuronal cells [10,11]. Such findings have led current models of sensory information coding in the brain to overlook the contribution of astrocytes, under the implicit or explicit assumption that astrocytic cells only provide information already encoded in neurons [13,14]. Here, we challenged this assumption and tested the hypothesis that astrocytes encode information in their intracellular calcium dynamics that is not present in the activity of nearby neurons. As a model, we used spatial information encoding in the

**Funding:** This work was funded by the European Research Council (https://erc.europa.eu/, NEURO-PATTERNS 647725) and NIH Brain Initiative (https://braininitiative.nih.gov/, U19 NS107464) to TF and National Institute of Health Brain Initiative (https://braininitiative.nih.gov/, U19 NS107464, R01 NS109961, R01 NS108410) to SP. The funders had no role in study design, data collection and analysis, decision to publish, or preparation of the manuscript.

**Competing interests:** The authors have declared that no competing interests exist.

**Abbreviations:** ACSF, artificial cerebrospinal fluid; COM, center of mass; DI, directionality index; DR, dynamic range; FOV, field of view; GECI, genetically encoded calcium indicator; GFAP, glial fibrillary acidic protein; MAD, median absolute deviation; ON, overnight; PB, phosphate buffer; PBS, phosphate buffered saline; PFA, paraformaldehyde; RF, response field; RP, response profile; ROI, region of interest; SD, standard deviation; SEM, standard error of the mean; SNR, signal-to-noise ratio; SP, spatial precision; SVM, support vector machine; t-series, temporal series.

hippocampus, where neural place cells encode navigational information by modulating their firing rate as a function of the animal's spatial location [15–17]. We demonstrate that astrocytic calcium signals encode information about the animal's position in virtual space and that, according to the statistical analysis we performed, this information is complementary to that carried by hippocampal neurons.

## Results

We combined two-photon functional imaging in head-fixed mice navigating in virtual reality [16,17] (Fig 1A) with astrocyte-specific expression of the genetically encoded calcium indicator GCaMP6f (Fig 1B and 1D, S1 Fig) [18–20]. To control for potential reactivity of astrocytes, we stained against the glial fibrillary acidic protein (GFAP) sections of fixed tissue from animals implanted with the chronic hippocampal window (S1 Fig). As internal controls, we used the contralateral nonimplanted hemisphere from the same experimental animals. We quantified GFAP signals in implanted and control hemispheres in three regions: the stratum Oriens, the stratum Pyramidale, and the stratum Radiatum. We found similar GFAP immunoreactivity in the stratum Pyramidale and Radiatum in implanted hemispheres compared to controls (S1E and S1F Fig). In contrast, we observed increased GFAP immunoreactivity in the stratum Oriens in implanted hemispheres compared to controls (S1E and S1F Fig). These results are in line with previous publications [21,22], which reported no astrocyte reactivity in the stratum pyramidale, where imaging was performed, and some astrocyte reactivity in a small region in the stratum Oriens close to the glass coverslip of the implant. We measured subcellular calcium dynamics of hippocampal CA1 astrocytes during spatial navigation in a virtual monodirectional corridor (Fig 1C) [23]. Using the intersection of two stringent criteria (significance of mutual information about spatial location carried by the cell's activity and reliability of calcium activity across running trials; Methods, S2 Fig), we found that a large fraction of astrocytic regions of interest (ROIs) had calcium signals that were reliably modulated by the spatial position of the animal in the virtual track (44 ± 21%, 155 out of 356 ROIs, from 7 imaging sessions on 3 animals, Fig 1E, S1 Table, S3 Fig). We defined the spatial response field of an astrocytic ROI as the portion of virtual corridor at which that ROI showed, on average across trials, increased GCaMP6f fluorescence (Methods). The distribution of astrocytic spatial response field positions covered the entire length of the virtual corridor (Fig 1F and 1G; $N$ = 155 ROIs from 7 imaging sessions on 3 animals). The median width of the astrocytic spatial field was 56 ± 22 cm ($N$ = 155 ROIs from 7 imaging sessions in 3 animals, Fig 1H). ROIs with reliable spatial information had reproducible estimates of spatial response profiles (S3B and S3C Fig). Splitting the dataset in odd and even trials resulted in a similar distribution of astrocytic field position compared to the entire dataset (Fig 1F center and rightmost panels, Fig 1I). We computed spatial precision as in [24] and found that calcium responses in astrocytic ROIs encoding reliable spatial information were moderately more precise than their unmodulated counterpart (S3D Fig; spatial precision, median ± MAD 3.2E-2 ± 0.6E-2, $N$ = 155 out of 356 total ROIs, for ROIs with reliable spatial information; 3.0E-2 ± 0.5E-2 cm$^{-1}$, $N$ = 201 out of 356 total ROIs, for not modulated ROIs: $p$ = 3.8E-2, Kolmogorov–Smirnov test; 7 imaging sessions on 3 animals). We computed response fields using running trials recorded either during the first or the second half of each experimental session. As in [24], we considered as stable those response fields showing an absolute difference in the estimated response field centers <15 cm. We found that a fraction of astrocytic ROIs (10 ± 10%, 35 out of 356 ROIs, from 7 sessions in 3 animals) encoded reliable spatial information and had stable response field. Moreover, we found that astrocytic calcium events were smaller when the mouse was still versus when the mouse was locomoting [25,26] and, for spatially modulated ROIs, in the absence versus presence of virtual reality (S4A and S4B Fig).

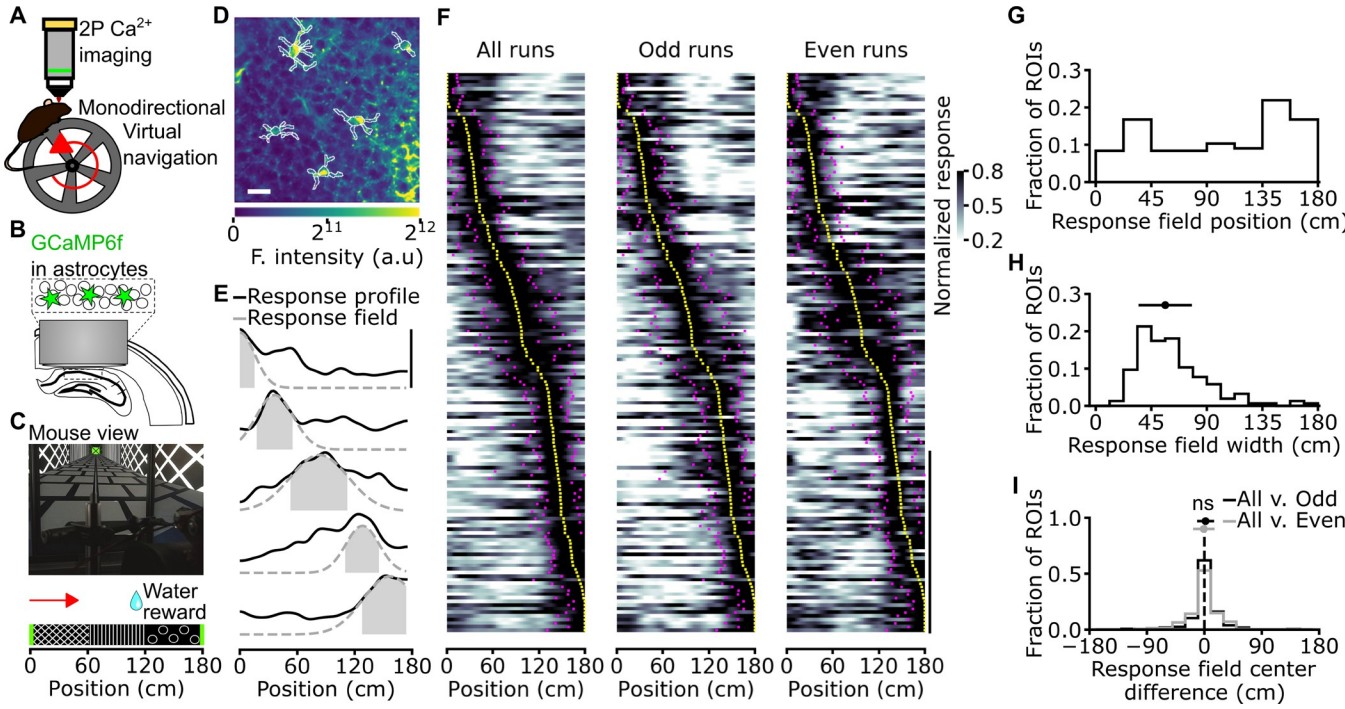

**Fig 1. Astrocytic calcium signals in the CA1 hippocampal area encode spatial information during virtual navigation.** (A) Two-photon fluorescence imaging was performed in head-fixed mice running along a monodirectional virtual track. (B) GCaMP6f was expressed in CA1 astrocytes, and imaging was performed through a chronic optical window. (C) Mice navigated in a virtual linear corridor in one direction, receiving a water reward in the second half of the virtual corridor. (D) Median projection of GCaMP6f-labeled astrocytes in the CA1 pyramidal layer. Scale bar: 20 µm. (E) Calcium signals for 5 representative astrocytic ROIs encoding spatial information across the corridor length. Solid black lines indicate the average astrocytic calcium response across trials as a function of spatial position. Dashed gray lines and filled gray areas indicate Gaussian fitting function and response field width (see Methods), respectively (see also S3 Fig). (F) Normalized astrocytic calcium responses as a function of position for astrocytic ROIs that contain significant spatial information ($N$ = 155 ROIs with reliable spatial information out of 356 total ROIs, 7 imaging sessions from 3 animals). Responses are ordered according to the position of the center of the response field (from minimum to maximum). Left panel, astrocytic calcium responses from all trials. Center and right panels, astrocytic calcium responses from odd (center) or even (right) trials. Yellow dots indicate the center position of the response field, and magenta dots indicate the extension of the field response (see Methods, vertical scale: 50 ROIs). (G) Distribution of response field position. (H) Distribution of field width. (I) Distribution of the differences between the center position of the response fields in cross-validated trials and odd trials (black) or cross-validated and even trails (gray). Deviations for odd and even trials are centered at 0 cm: median deviation for odd trials 2 ± 13 cm; median deviation for even trials −1 ± 17 cm, neither is significantly different from zero ($p$ = 0.07 and $p$ = 0.69, respectively, Wilcoxon signed rank test with Bonferroni correction. $N$ = 155 ROIs from 7 imaging sessions on 3 animals). The data presented in this figure can be found in S1 Data. ROI, region of interest.

Experiments performed with mice trained in a bidirectional virtual corridor (S5 Fig) [16,17] confirmed the results obtained in the monodirectional virtual corridor: a significant fraction of astrocytic ROIs carried significant information about the spatial position of the animal in the virtual corridor and the distribution of positions of the astrocytic spatial field covered the whole virtual corridor (29 ± 13%, $N$ = 192 out of 648 ROIs in the forward direction; 20 ± 13%, $N$ = 133 out of 648 ROIs in the backward direction, $p$ = 0.09 Wilcoxon signed rank test for comparison between forward and backward directions, from 18 imaging sessions in 4 animals; S5E and S5F Fig). The median width of the spatial response field was 44 ± 20 cm, $N$ = 192 out of 648 ROIs in the forward direction and 44 ± 29 cm, $N$ = 133 out of 648 ROIs in the backward direction ($p$ = 0.34 Wilcoxon rank sums test for comparison between forward and backward directions, S5G Fig). In the bidirectional virtual corridor, astrocytic ROIs showed significant direction selective spatial modulation in their response field (S5H Fig). Thus, in the hippocampus astrocytic calcium signaling encoded spatial information.

Astrocytic calcium signaling has been shown to be organized at the subcellular level; the calcium dynamics of astrocytic cellular processes can be distinct from those occurring in the

astrocytic cell body [2,3,5,7,8]. We thus categorized astrocytic ROIs (among the set of 356 described above) according to whether they were located within main processes (process ROIs) or cell bodies (soma ROIs, Fig 2). Signals from both soma ROIs and process ROIs encoded spatial information (Fig 2A). Moreover, a similar fraction of soma ROIs and process ROIs were modulated by the spatial position of the animal (42 ± 34%, 19 out of 46 soma ROIs versus 44 ± 21%, 136 out of 310 process ROIs, $p$ = 0.61 Wilcoxon signed rank test, from 7 imaging sessions on 3 animals). The distribution of field position of soma ROIs and process ROIs similarly covered the entire length of the virtual corridor (Fig 2B, S6A Fig, S1 Table). The width of the astrocytic spatial field did not differ between process ROIs and soma ROIs (S6B Fig). Within individual astrocytes, the difference between the field position of a process ROI and the corresponding soma ROI (both containing reliable spatial information) increased as a function of the distance between the two ROIs (Fig 2C, S6 Fig). Thus, spatial information was differentially encoded in topographically distinct locations of the same astrocyte. The difference between the field position of a process ROI and the corresponding soma ROI did not depend on the angular position of the process with respect to the soma (S6 Fig). When comparing calcium activity across pairs of ROIs with reliable spatial information (belonging to processes or somas across astrocytes), correlation decreased as a function of the pair distance ($\tau_{decay}$ = 14 ± 2 μm, $R^2$ = 0.98) in the 0- to 50-μm range and then substantially plateaued for pair distances between 50 μm and 160 μm (Fig 2F). This indicates that calcium signals encoding reliable spatial information were coordinated across distant ROIs, even those putatively belonging to different cells. In agreement with this observation, the difference in field position among pairs of ROIs with reliable spatial information increased as a function of pair distance within 0 to 40 μm and then plateaued to a constant value ($\tau_{rise}$ = 13 ± 7 μm, $R^2$ = 0.79) for pair distances between 40 μm and 160 μm (Fig 2G). Event triggered averages of astrocytic responses representing temporal relationships between calcium signals at different subcellular regions are shown in S7 Fig.

Since calcium dynamics of individual astrocytic ROIs encodes significant spatial information, it should be possible to decode the animal's position in the virtual corridor from single-trial calcium dynamics of populations of astrocytic ROIs. We trained a support vector machine (SVM) to classify the mouse's position according to a set of discrete spatial locations using a single-trial population vector made combining calcium signals of all individual astrocytic ROIs within the field of view (FOV). We computed the population decoding accuracy and the decoded spatial information [27] as a function of spatial granularity, i.e., the number of discrete locations available to the SVM decoder (4, 8, 12, 16, 20, or 24 locations). We found that the SVM predicted the animal's spatial location across granularities (Fig 3A, S1 Table). Cross-validated decoding accuracy (S8 Fig) and decoded spatial information were significantly above chance (Fig 3B) across the entire range of spatial granularities (chance level was estimated by decoding position after randomly shuffling spatial locations in the data while preserving the temporal structure of the population calcium signals; see Methods). Disrupting the within-trial temporal coupling within astrocytic population vectors while preserving single ROI activity patterns [28,29] consistently decreased decoded spatial information (Fig 3B) and decoding accuracy (S8 Fig). This suggests that within-trial interactions among astrocytic ROIs encode spatial information not present in their individual activities. Misclassifications were more likely to happen among nearby locations across all granularity conditions (Fig 3C), consistent with the idea that astrocytic activity allows localization of the animal's position. Experiments performed with mice trained in a bidirectional virtual environment (S9 Fig) largely confirmed these decoding results.

Our virtual corridor was characterized by the alternation of three different patterns (grid, vertical lines, and circles) similarly to [16,17,24]. The three patterns covered the whole length

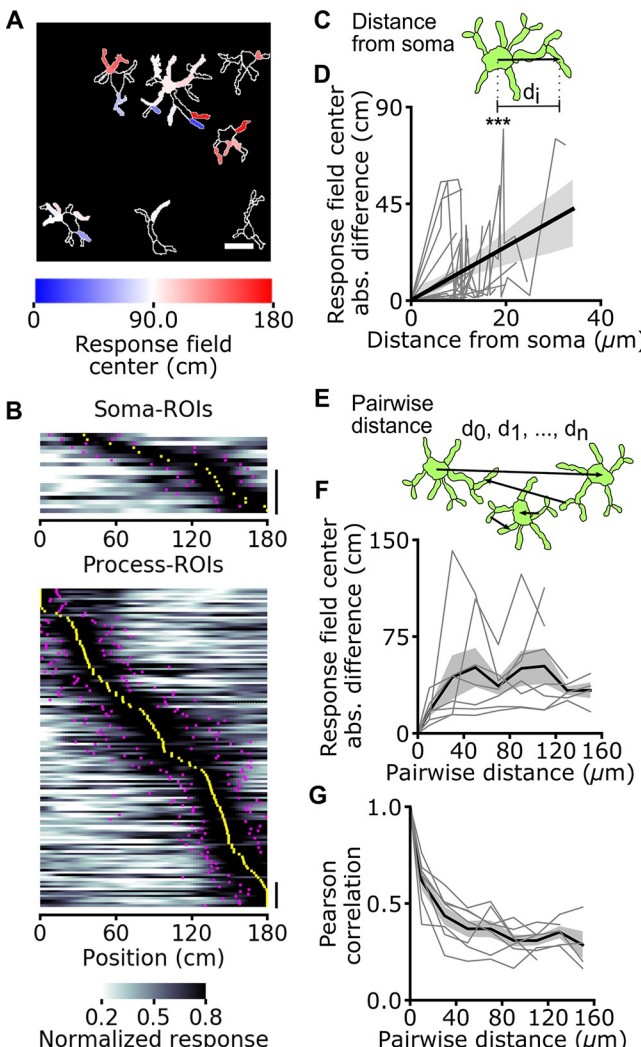

**Fig 2. Topographic organization of spatial information encoding in astrocytes: somas versus processes. (A)** Astrocytic ROIs in a representative FOV are color coded according to response field position along the virtual corridor. Scale bar: 20 μm. **(B)** Normalized astrocytic calcium responses as a function of position for astrocytic ROIs with reliable spatial information corresponding to somas (top) and processes (bottom) (somas: 19 ROIs with reliable spatial information out of 46 total ROIs; processes: 136 ROIs with reliable spatial information out of 310 total ROIs; data from 7 imaging sessions in 3 animals). Vertical scale: 10 ROIs. **(C)** Distance between the center of a process ROI and corresponding soma ROI computed for each astrocyte. **(D)** Absolute difference in response field position of a process ROI with respect to the field position of the corresponding soma ROI as a function of the distance between the 2 ($R^2 = 0.21$, $p = 3.2\text{E-}6$, Wald test, data from 19 cells in which there was significant spatial modulation in the soma and at least 1 process; 7 imaging sessions on 3 animals). **(E)** The distance between the centers of pairs of ROIs ($d_0$, $d_1$, $d_n$) is computed across recorded astrocytic ROIs. **(F, G)** Pearson correlation (F) and difference between response field position (G) for pairs of astrocytic ROIs containing reliable spatial information across the whole FOV as a function of pairwise ROI distance. Gray lines indicate single experiments, and black line and the gray shade indicate mean ± SEM, respectively. Data from 41 cells in which there was significant spatial modulation in at least 1 ROI; 7 imaging sessions in 3 animals. In this as well as in other figures: *, $p < 0.05$; **, $p \leq 0.01$; ***, $p \leq 0.001$. The data presented in this figure can be found in S1 Data. FOV, field of view; ROI, region of interest; SEM, standard error of the mean.

of the virtual corridor (180 cm) and each pattern was presented for 60 cm of the corridor. Within each of these 60 cm–long visual cues, the visual stimuli associated with each pattern were periodically repeated (Fig 1C). Can the different visual cues account for the modulation of spatial information that we observed in astrocytes? We reasoned that if astrocytes responses

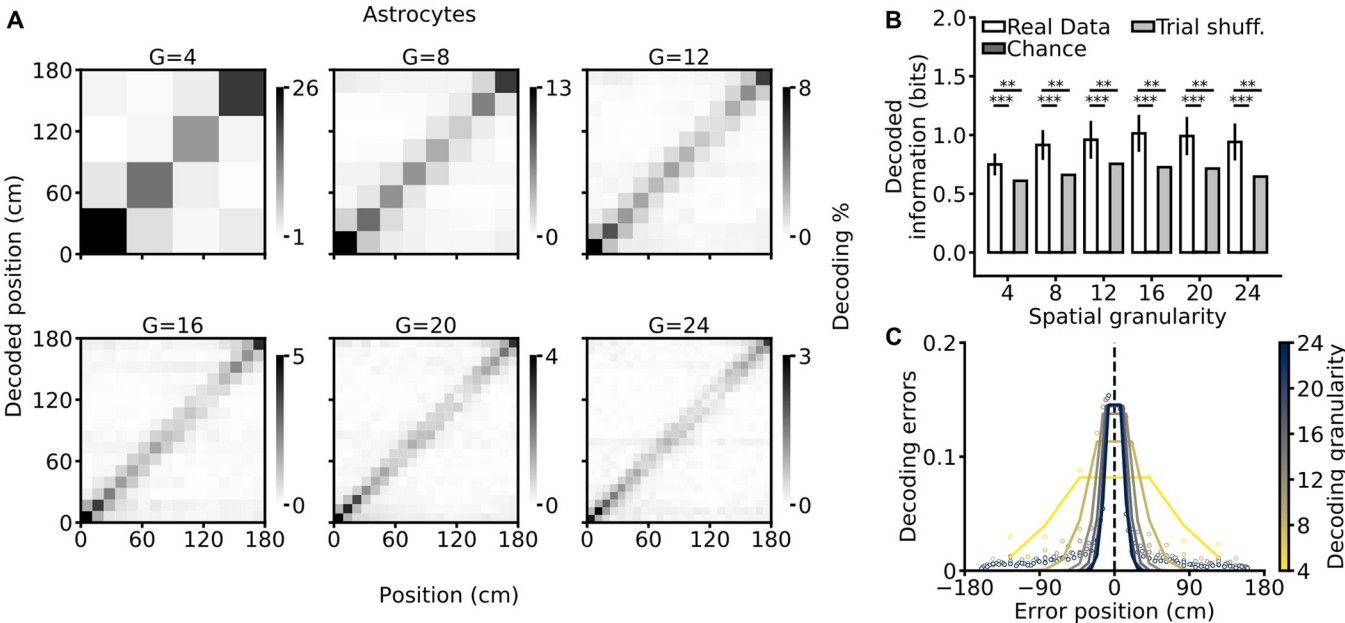

**Fig 3. Efficient decoding of the animal's spatial location from astrocytic calcium signals. (A)** Confusion matrices of an SVM classifier for different decoding granularities (G = 4, 8, 12, 16, 20, and 24). The actual position of the animal is shown on the x-axis, and decoded position is on the y-axis. The gray scale indicates the number of events in each matrix element. **(B)** Decoded information as a function of decoding granularity on real (white), chance (dark gray), and trial-shuffled (gray) data (see Methods). Trial shuffling disrupts temporal coupling within astrocytic population vectors while preserving single ROI activity patterns. Data are shown as mean ± SEM. See also S2 Table. **(C)** Decoding error as a function of the error position within the confusion matrix. The color code indicates decoding granularity. Data in all panels were obtained from 7 imaging sessions in 3 animals. The data presented in this figure can be found in S1 Data. ROI, region of interest; SEM, standard error of the mean; SVM, support vector machine.

in the virtual reality corridor were only modulated by visual cues regardless of the position in which the visual stimulus was provided, then astrocytes calcium responses should not have the power to discriminate between spatial locations within the 60 cm–long spatial interval in which a single visual cue was presented. In such case, the astrocytic responses would not carry spatial information above and beyond the one that is inherited from the information they carry about the identity of the visual cue. To test whether astrocytic signals carried spatial information that cannot be possibly attributed to visual cue modulation, we randomly shuffled the relationship between position and astrocytic signals within each visual cue. This data shuffling procedure preserves cue information carried by the astrocytes but destroys all the genuine spatial information they carry above and beyond visual cue information. The difference between the information carried by the real, unshuffled, responses and the information carried by the shuffled responses quantifies the amount of spatial position information carried by the astrocytes that cannot be possibly attributed to spatial cue tuning. Analyzing individual astrocytic ROIs (Fig 4A), we found that a large fraction (approximately 50% to 60%) of spatially modulated ROIs carried significantly more information than what could be solely explained by visual cue identity. Moreover, when decoding the animals' position from astrocytic population vectors (Fig 4B and 4C, S10 Fig), we found that the majority (approximately 55% to 65%) of the decoded information was genuinely information about position. We performed both analysis dividing each visual cue in a number of spatial bins that was systematically varied from 3 to 6, leading to an overall spatial granularity varying from 9 to 18, and obtaining qualitatively similar results across granularities.

How does the astrocytic representation of spatial information relate to that of neuronal cells? We combined astrocyte-specific expression of GCaMP6f with neuronal expression of

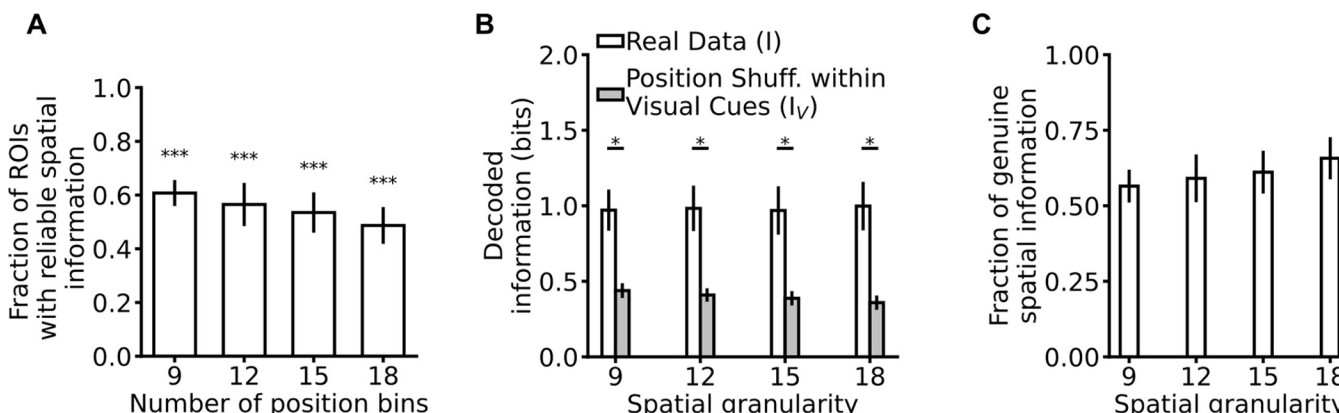

**Fig 4. The majority of spatial information in astrocytes is genuine spatial information that cannot be explained by tuning to visual cues. (A)** Fraction of astrocytic ROIs encoding reliable spatial information showing a significant decrease in their information content when position is shuffled within the same visual cue (see Methods). Shuffling position within the same visual cue decouples spatial information encoded in the astrocytic response from the information related to visual cues identity (see Methods). The fraction of ROIs showing significant information loss is shown as function of the number of position bins used to compute mutual information. $p$ = 3.5E-168, $p$ = 3.2E-138, $p$ = 5.0E-133, and $p$ = 5.2E-85 for 9, 12, 15, and 18 position bins, respectively; $N$ = 155, binomial test. **(B)** Decoded information as a function of decoding granularity on real data (I, white) and for data in which position is shuffled within the same visual cue ($I_V$, gray). $p$ = 1.6E-2, $p$ = 1.6E-2, $p$ = 1.6E-2, and $p$ = 1.6E-2 for decoding granularity of 9, 12, 15, and 18, respectively. $N$ = 7 imaging sessions, Wilcoxon signed rank test. See also S10 Fig. **(C)** Fraction of genuine spatial information in astrocytic population vectors computed shuffling position within individual visual cues. Results are shown as a function of decoding granularity. In all panels, data are shown as mean ± SEM and were obtained from 7 imaging sessions in 3 animals. The data presented in this figure can be found in S1 Data. ROI, region of interest; SEM, standard error of the mean.

jRCaMP1a [30] and performed simultaneous dual color hippocampal imaging with two-photon microscopy (Fig 5A and 5B, S11 Fig) during virtual navigation. We found that a sizable fraction of astrocytic and neuronal ROIs (astrocytes, 22 ± 19%, 76 out of 341 ROIs; neurons, 38 ± 13%, 335 out of 870 ROIs, from 11 imaging sessions on 7 animals) reliably encoded information about the spatial position of the animal in the virtual corridor. For both astrocytes and neurons, the distribution of field position covered the entire length of the virtual corridor (Fig 5C and 5D). The median width of the astrocytic spatial field was statistically larger than that of neurons (Fig 5E, S1 Table). Event triggered averages of astrocytic ROIs signals triggered by neuronal signals are shown in S12 Fig. Both neuronal and astrocytic calcium events were bigger when the mouse was engaged in locomotion (S13A Fig) and for spatially modulated ROIs in the presence versus the absence of virtual reality (S13B Fig). We then investigated the organization of astrocytic and neuronal spatial representations across the FOV. We found that calcium dynamics among mixed pairs of ROIs (one astrocytic ROI with reliable spatial information and one neuronal ROI with reliable spatial information) were significantly correlated (S14 Fig), independently of pair distance (0 to 160 μm; Fig 5F). Correlation among pairs of astrocytic ROIs was generally higher than correlation among pairs of neuronal ROIs (S14 and S15 Figs), even when we stratified the calculation of pairwise correlation for pairs of ROIs belonging to the same astrocyte and for pairs of ROIs belonging to different astrocytes (S14 Fig). The difference in spatial field position of an astrocytic ROI with reliable spatial information and a neuronal ROI with reliable spatial information was also largely independent of pair distance (Fig 5G). We compared the spatial precision [24] of astrocytic responses with that of neuronal responses. We found that the responses of position-encoding neurons were more precise than the responses of simultaneously recorded position-encoding astrocytic ROIs (mean ± SEM; neuronal responses 7.5E-2 ± 1.6E-2; astrocytic responses 4.1E-2 ± 0.2E-2; $p$ = 4.6E-2 Wilcoxon signed rank test; 11 imaging sessions on 7 animals, S16A Fig). We also compared astrocytic response field stability with neuronal place field stability. We found that similar fractions of astrocytic ROIs and neuronal cells encoded reliable spatial information

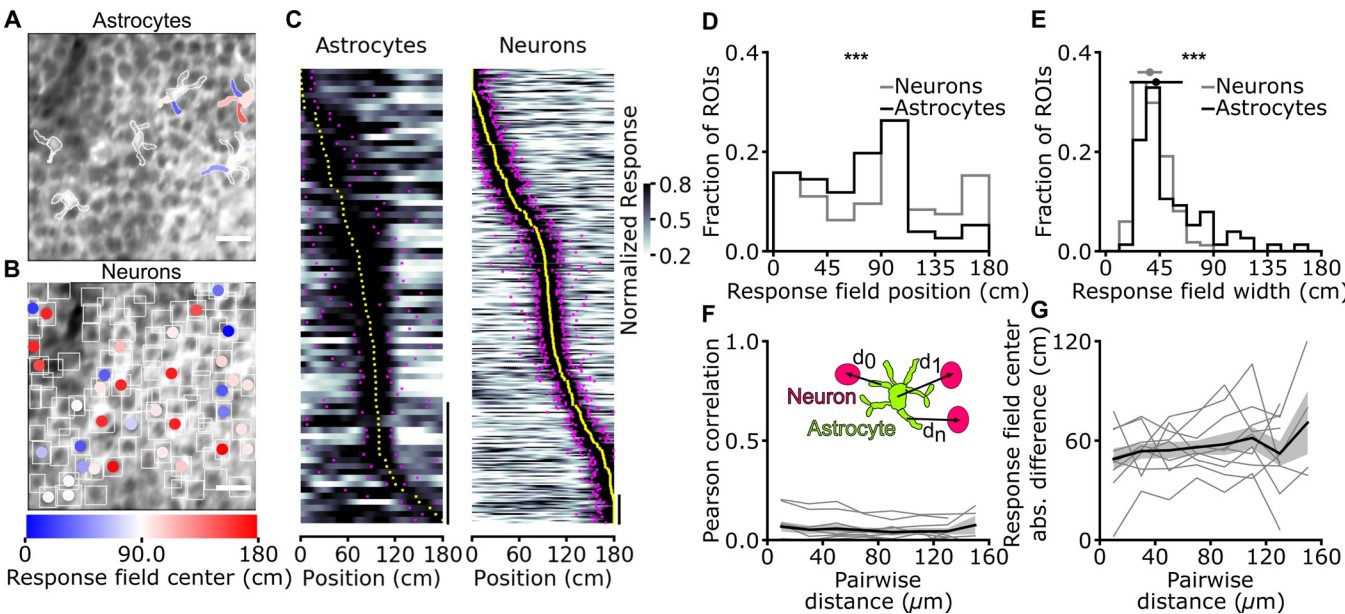

**Fig 5. Astrocytes have broader response field width and a different distribution of field position compared to neurons. (A, B)** ROIs corresponding to simultaneously recorded GCaMP6f-labeled astrocytes (A) and jRCaMP1a-labeled neurons (B) in the CA1 pyramidal layer. ROIs are color coded according to response field and place field center along the virtual corridor, respectively. Scale bar: 20 μm. **(C)** Normalized calcium responses as a function of position for astrocytic ROIs (left) and neuronal ROIs (right) that contain a significant amount of spatial information (astrocytic ROIs, $N = 76$ ROIs with reliable spatial information out of 341 total ROIs; neuronal ROIs, $N = 335$ ROIs with reliable spatial information out of 870 total ROIs, data from 11 imaging sessions in 7 animals). Responses are ordered according to the position of the center of the response field for astrocytes and place field for neurons. Vertical scale bar, 20 ROIs. **(D)** Distribution of astrocytic response field position (black line) and neuronal place field position (gray line, $p = 5E-4$, Kolmogorov–Smirnov test for comparison between astrocytic and neuronal distribution). **(E)** Distribution of astrocytic response field width (black line) and neuronal place field width (gray line, median width of astrocytic response field: 42 ± 22 cm, $N = 76$; median width of neuronal place field: 37 ± 10 cm, $N = 335$, $p = 2E-5$, Wilcoxon rank sums test for comparison between astrocytic and neuronal distribution). **(F, G)** The inset shows astrocytic ROIs (green) and neuronal ROIs (pink). For all pairs, the distance ($d_0$, $d_1$, $d_n$) between the center of an astrocytic ROI and the center of a neuronal ROI, both containing reliable spatial information, is computed. Pairwise Pearson correlation (F) and difference between response field position for astrocyte–neuron ROI pairs (G) as a function of pair distance. In (F,G) Data are expressed as mean ± SEM. Data are from 11 imaging sessions in 7 animals (see also S15 Fig). The data presented in this figure can be found in S1 Data. ROI, region of interest; SEM, standard error of the mean.

and had stable response field (astrocytes, 8 ± 7%, 29 out of 341 ROIs; neurons, 16 ± 9%, 139 out of 870 ROIs; $p = 0.29$, Wilcoxon rank sums test; from 11 imaging sessions on 7 animals, S16B Fig). Importantly, we also found that a large fraction of astrocytic and neuronal ROIs showing spatial modulation carried a significant amount of spatial information that could not be explained by visual cue tuning (S17A and S17B Fig). Moreover, when analyzing population vectors using an SVM decoder, the majority (approximately 60% to 80%, S18 Fig) of the total spatial information carried by either astrocytic or neuronal ROIs could not be possibly explained by visual cue modulation. Thus, the majority of spatial information in astrocytes and neurons is genuine spatial information that cannot be explained by tuning to visual cues.

We then quantitatively tested whether calcium dynamics in astrocytes and neurons carry the same or complementary information about space. We did so at the pairwise level using mutual information analysis [27] on all pairs of ROIs (either astrocytic, neuronal, or mixed pairs). Regardless of pair identity, we found that information carried by pairs of ROIs was greater than information carried by either ROI individually (Fig 6A, S19 Fig). Moreover, information carried by pairs of ROIs was higher than the sum of the information carried by each of 2 ROIs, regardless of pair identity (Fig 6A, S19 Fig, S1 Table). Thus, information carried by the pairs was also synergistic. To understand how correlations between ROIs leads to synergistic coding, we used mutual information breakdown analysis of ROI pairs [31,32]. This revealed

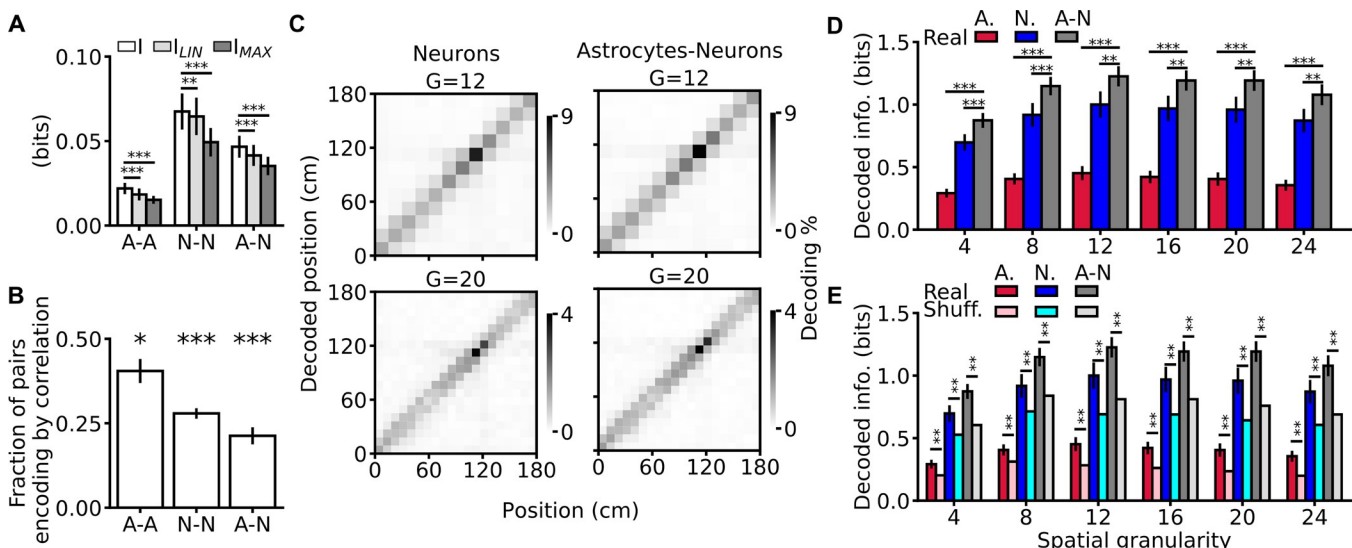

**Fig 6. Spatial information encoding in astrocytes is complementary and synergistic to spatial information encoding in neurons. (A)** Information about position carried by pairs of ROIs (I) compared to the sum ($I_{LIN}$) or the maximum ($I_{MAX}$) of the information separately encoded by each member of the pair. A-A, pair composed of 2 astrocytic ROIs; N-N, pair composed of 2 neuronal ROIs; A-N, mixed pair composed of one astrocytic and one neuronal ROI (I versus $I_{LIN}$: A-A: $p$ = 1E-3, N-N: $p$ = 5E-3, A-N: $p$ = 1E-3; I versus $I_{MAX}$: A-A: $p$ = 1E-3, N-N: $p$ = 1E-3, A-N: $p$ = 1E-3, Wilcoxon signed rank test, see also S19 Fig and S3 Table). **(B)** Fraction of pairs encoding spatial information encoding by correlations (A-A: $p$ = 3E-2, N-N: $p$ = 1E-3, A-N: $p$ = 1E-3, Wilcoxon signed rank-test with respect to the null hypothesis that a pair could be either synergistic or nonsynergistic with equal probability set at 0.5). **(C)** Representative confusion matrices of an SVM classifier decoding mouse position using population vectors comprising neuronal (left) or astrocytic and neuronal ROIs (right), for different decoding granularities (G = 12, 20, see also S21 Fig). **(D)** Decoded information for population vectors of different compositions (A, astrocytic ROIs only; N, neuronal ROIs only; A-N, population vector considering all ROIs) as a function of decoding granularity (see S4 Table). **(E)** Same as in (D) but adding comparison with trial-shuffled data (lighter bars) (see S5 Table). In panels A, B, D, and E, data are represented as mean ± SEM. In all panels, data are obtained from 11 imaging sessions in 7 animals. The data presented in this figure can be found in S1 Data. ROI, region of interest; SEM, standard error of the mean; SVM, support vector machine.

two notable results. First, the "signal similarity" component of information ($I_{SS}$), which quantifies the reduction of ROI pair information, or redundancy, due to the similarity of the trial averaged response profiles of the individual ROIs (see Methods and S20 Fig), was close to zero. Thus, the diversity of spatial profiles allowed ROIs to sum up their information with essentially no redundancy. Second, synergy between elements of pairs was based on a positive stimulus-dependent correlation component ($I_{CD}$, see Methods and S20 Fig), which contributed to increase the joint information. Mathematically, $I_{CD}$ can be nonzero if and only if within-trial correlations between ROIs are modulated by the animal's position and they carry information complementary to that given by position modulation of each individual ROI [32]. Correlation enhancement of spatial information was found in a sizeable fraction of pairs across all pair identities, including mixed pairs (Fig 6B). This was because the strength of correlations between neurons and astrocytes marked the position in virtual corridor: for pairs of one neuronal ROI and one astrocytic ROI, the absolute magnitude of correlations showed a position-dependent modulation (S21 Fig), with stronger correlations inside the spatial fields.

Complementary and synergistic spatial information encoding in mixed pairs suggested that the network of astrocytes that we imaged carried spatial information that was not found in the imaged neurons and in their interactions. To directly address this hypothesis, we computed the spatial information gained by decoding the animals' position from an SVM operating on population vectors comprising either all neuronal, all astrocytic, or all ROIs of both types. We found that neuronal, astrocytic, and mixed population vectors allowed to classify the animal's position across granularity conditions (Fig 6C–6E, S22 Fig). However, decoding population vectors comprising both astrocytic and neuronal ROIs led to a greater amount of spatial

information than decoding either neuronal or astrocytic population vectors separately (Fig 6D). This result supports the hypothesis that the population of astrocytic ROIs encodes information not found in neurons or their interactions. In agreement with what we found in the pair analysis, information decoded from all types of population vectors decreased when within-trial temporal correlations between cells were disrupted by trial shuffling (Fig 6E, S22 Fig) [28,32]. Within-trial correlations were thus an important factor for the complementary and synergistic contribution of astrocytes to spatial information encoding at the population level.

## Discussion

Our findings demonstrate, for the first time, that information-encoding cellular signals during virtual spatial cognition extend beyond neuronal circuits to include the nearby astrocytic network. This information was expressed in spatially restricted subcellular regions, including cellular processes and somas, in agreement with previous work describing the complexity and compartmentalization of calcium signals in these glial cells [2–5,8,33]. Importantly, individual astrocytes could encode multiple spatial fields across different subcellular compartments, suggesting that a single astrocyte may integrate multiple neuronal spatial representations. Interestingly, the spatial representations in individual astrocytes displayed a concentric organization: the difference between the place field position of a subcellular process and the place field position of the corresponding soma increases as a function of distance. Most importantly, combining astrocytic and neuronal signals generated significantly greater information about the animal's position, suggesting that the signals are both complementary and synergistic. The complementary and synergistic information of astrocytes relied both on the diversity of position tuning and on position-dependent correlations among astrocytic and neuronal ROIs similarly to what observed on neuronal ROIs by [29]. It should also be considered that astrocytic signals may convey complementary information by simultaneously integrating the activity of several different neuronal inputs encoding distinct stimulus-related variables [34–36].

Since the seminal observations of Cornell-Bell and colleagues [37] and Charles and colleagues [38], it has been shown that astrocytic calcium signaling can be activated by the extracellular increase in the concentration of neuroactive molecules [1,39,40]. The source of the observed calcium signaling has long been investigated, and there is evidence of both release from intracellular organelles (e.g., endoplasmic reticulum and mitochondria) [6,41] and direct calcium influx through the astrocytic plasma membrane [42]. Moreover, while neuronal activity induces membrane depolarization in astrocytes, these depolarizations are small (a few millivolts in maximal amplitude), at least at the level of the astrocytic somata [2,43,44]. Pharmacological studies demonstrated that these membrane potential depolarizations are mediated mostly by $K^+$ conductances and transporters-mediated currents [45]. For neuronal activity–induced calcium signals, a largely accepted model [39, 40] is thus that spillover of neurotransmitter release at the synaptic cleft activates receptors in the plasma membrane of astrocytic processes, which enwrap the pre- and the postsynaptic terminals. Activation of astrocytic receptors then causes the mobilization of intracellular calcium signaling. Within this framework, it is interesting to note that previous studies in vivo showed that calcium dynamics into astrocytes largely mirror the activity of nearby neurons [7,10,11]. The information theoretic approach we used in the present study instead shows that spatial information encoded in astrocytes is complementary to that encoded by nearby neurons. This observation raises a series of questions about the molecular, anatomical, and network mechanisms that may generate the observed information complementarity. Can complementarity be generated by differences in the diffusion of presynaptically released neurotransmitter that reaches postsynaptic neuronal terminals in the synaptic cleft versus thin astrocytic processes outside the cleft? Are the

different molecular mechanisms regulating intracellular signaling in neurons and astrocytes accountable for the observed information complementarity? Additionally, can information complementarity stem from astrocytes integrating spatial information from different sets (or different combinations) of presynaptic terminals compared to postsynaptic neurons? Our work does not directly address the questions raised above and future studies combining experimental and modeling work will be fundamental to tackle these questions. It is important to note that demonstrating by statistical analysis of cell activity, as we did here, that astrocytes carry complementary spatial information is not sufficient to demonstrate that this information is causal to circuit function or behavior. Experimental manipulations targeted to either astrocytes or neurons must be used to establish causality [46]. To this aim, it will thus be necessary to perturb hippocampal astrocytic and neuronal networks with high spatial and temporal resolution [47–49] while monitoring downstream effects on circuit functions and behavior.

Models of hippocampal function posit that information about variables of the external environment, which are key to spatial navigation and memory, is exclusively encoded in population of neurons [50–53]. Our results challenge this established view by revealing a fundamental new level of organization for information encoding in the hippocampus during virtual navigation: spatial information, which according to the information theoretical analysis used in this study is not available in the activity of CA1 projecting neuron or in their interactions, is encoded in the calcium dynamics of local nonneuronal elements and in their position-dependent interaction with neurons. The presence of this additional nonneural reservoir of information and the dependence of the interaction between neuron and astrocytes on key cognitive variables suggest the possible presence of novel and unexpected cellular mechanisms underlying how brain circuits encode information.

Can complementary spatial information encoded in astrocytic calcium dynamics contribute to neuronal computation? If so, how? Although our data do not address these questions, previous work in other brain regions reported that astrocytic calcium dynamics largely mirror the activity of nearby neurons [7,10,11] and that astrocytic signals translate into changes in neuronal excitability and synaptic transmission through various mechanisms (reviewed in [1,54–56]). In this scenario, changes in synaptic transmission and neuronal excitability induced by astrocytic calcium dynamics that simply mirror neuronal information would only modulate neural activity without providing further information, as all the activity-dependent information is already encoded in the neuronal activity. For example, if the neuronal tuning curve and the astrocytic-induced change in neural function are similarly modulated by the animal's position, no additional dependence of neuronal function by position would be introduced by astrocyte–neuron interactions. Conversely, our findings suggest that astrocytic calcium dynamics carrying complementary information to that of neurons enable modulations of synaptic transmission and neuronal gain that could increase the computational capability of neural circuits [57,58]. For example, changing the gain of neurons with a coordinate system complementary to that regulating its tuning function has been shown to endow neural networks with richer computations [58,59]. Moreover, targeted dynamic control of neural excitability (e.g., changing the gain of a subset of neurons in the network rather than the whole network) can greatly increase the dynamic repertoire and coding capabilities of circuits, for example, by making it possible to reach different attractors from a similar set of initial conditions [60]. We thus propose that the complementary place dependence of the astrocytic calcium dynamics and the place dependence of astrocytic–neuron interactions reported here facilitate the emergence of dynamic, context-dependent changes in population coding of CA1 neurons. Within this scenario, local neuromodulation of the space-encoding properties of projecting CA1 neurons by astrocytes could affect hippocampal output. Future experiments involving selective perturbation (e.g., activation or inactivation) of astrocytic calcium signaling

will be needed to test this hypothesis. Our work calls for a reexamination of the theory of place coding and of brain population codes in light of the opportunities offered by the suggested complementary astrocytic information coding. We propose that the complementary regulation of astrocytic calcium activity and of its interaction with neurons may reflect a general principle of how the brain encodes information. This conclusion may extend beyond the hippocampus and spatial navigation to other brain regions and cognitive tasks, and it will need to be included in the conceptualization of brain function.

# Methods

## Animals

All experiments involving animals were approved by the National Council on Animal Care of the Italian Ministry of Health (authorization 61/2019-PR) and carried out in accordance with the guidelines established by the European Communities Council Directive. From postnatal day 30, animals were separated from the original cage and housed in groups of up to 5 littermates per cage with ad libitum access to food and water in a 12-hour light–dark cycle. Experimental procedures were conducted on animals older than 10 weeks. The number of animals used for each experimental dataset is specified in the text or in the figure legends.

## Adeno-associated virus (AAV) injection and chronic hippocampal window surgery

Astrocytic-specific GCaMP6f expression was obtained using pZac2.1 gfaABC1D-cyto-GCaMP6f (Addgene viral prep # 52925-AAV5, a gift from Dr. Khakh [4, 20]). Neuronal-specific jRCaMP1a expression was obtained using pAAV-CAMKII-jRCaMP1a (kindly provided by Dr. O. Yizhar), which was then packaged as AAV serotype 1–2 viral particles [61].

Male C57Bl6/j mice were anesthetized with 2% isoflurane/0.8% oxygen, placed into a stereotaxic apparatus (Stoelting, Wood Dale, Illinois, United States of America) and maintained on a warm platform at 37˚C for the whole duration of the anesthesia. Before surgery, a bolus of Dexamethasone (4 mg/kg, Dexadreson, MSD Animal Health, Milan, Italy) was provided with an intramuscular injection. After scalp incision, a 0.5-mm craniotomy was drilled on the right hemisphere (1.75 mm posterior and 1.35 mm lateral to bregma), and the AAV-loaded micropipette was lowered into the hippocampal CA1 region (1.40 mm deep to bregma). 800 nL of AAV solution was injected at 100 nL/min by means of a hydraulic injection apparatus driven by a syringe pump (UltraMicroPump, World Precision Instruments, Sarasota, Florida, USA). Following the viral injection, a stainless-steel screw was implanted on the cranium of the left hemisphere and a chronic hippocampal window was implanted similarly to [17,24]. A drill was used to open a 3-mm craniotomy centered at coordinates 2.00 mm posterior and 1.80 mm lateral to bregma. The dura was removed using fine forceps, and the cortical tissue overlaying the hippocampus was carefully aspirated using a blunt needle coupled to a vacuum pump. During aspiration, the exposed tissue was continuously irrigated with HEPES-buffered artificial cerebrospinal fluid (ACSF). Aspiration was stopped once the thin fibers of the external capsule were visible. A cylindrical cannula-based optical window was fitted to the craniotomy in contact to the external capsule and a thin layer of silicone elastomer (Kwik-Sil, World Precision Instruments) was used to surround the interface between the brain tissue and the steel surface of the optical window. A custom stainless-steel headplate was attached to the skull using epoxy glue. All the components were secured in place using black dental cement, and the scalp incision was sutured to adhere to the implant. Animals received an intraperitoneal bolus of antibiotic (BAYTRIL, Bayer, Germany) at the end of the surgery.

Optical windows were composed of a thin-walled stainless-steel cannula segment (OD, 3 mm; ID, 2.77 mm; height, 1.50 to 1.60 mm). A 3.00-mm diameter round coverslip was attached to one end of the cannula using UV curable optical epoxy (Norland optical adhesive 63, Norland, Cranbury, New Jersey, USA). Sharp edges and bonding residues were smoothed using a diamond-coated cutter.

## Virtual reality

A custom virtual reality setup was implemented using the open-source 3D creation suite Blender (blender.org, version 2.78c). Virtual environment rendering was performed using the Blender Game Engine and displayed at video rate (60 Hz). The virtual environment was a linear corridor with the proximal walls characterized by 3 different white textures (vertical lines, mesh, and circles) on a black background. Distal walls were colored in green and labeled with a black cross. The corridor was 180 cm long and 9 cm wide. The character avatar was a sphere of radius 2 cm with a rectangular cuboid protruding at the equator parallel to the corridor floor (cuboid dimension: x = 5 cm, y = 1 cm, z = 1 cm). The cuboid acted as a virtual touch sensor with the environment. The character point of view (220˚ horizontal and 80˚ vertical) was rendered through a composite tiling of 5 thin bezel-led screens. The virtual corridor implementation described above was used for both monodirectional and bidirectional navigation. In monodirectional virtual navigation, mice navigated the environment running on a custom 3D printed wheel (radius 8 cm and width 9 cm). An optical rotary encoder (Avago AEDB-9140-A14, Broadcom, San Jose, California, USA) captured motion and a single board microcontroller (Arduino Uno R3, Arduino, Ivrea, Italy) performed USB-HID-compliant conversion to a serial mouse input. In bidirectional virtual navigation, mice navigated the environment using an air-suspended Styrofoam ball (radius, 10 cm), and a Bluetooth optical mouse (M170, Logitech, Lausanne, Switzerland) was used to read the vertical and horizontal displacement. In both monodirectional and bidirectional navigation, physical motion of the input devices was mapped 1:1 to the virtual environment. To motivate corridor navigation, mice received approximately 4-μl water rewards upon reaching specific locations. Rewards were delivered through a custom steel lick-port controlled by a solenoid valve (00431960, Christian Bürker, Ingelfingen, Germany), and licks were monitored using a capacitive sensor (MTCH102, Microchip Technology, Chandler, Arizona, USA). In monodirectional virtual navigation, rewards were delivered at 115 cm and the mouse was teleported to the beginning of the corridor after reaching the end of the track (intertrial timeout interval 5 seconds). If the mouse did not reach the end of the corridor within 120 seconds, the trial was automatically terminated, and the mouse was teleported to the beginning of the corridor after an intertrial timeout. For bidirectional navigation, rewards were delivered at opposite ends of the track. After getting a reward, the mouse had to reach the opposite end of the virtual corridor to receive the next reward. Virtual reality rendering and 2-photon imaging acquisition ran on asynchronous clocks, while the command signal of the slow galvanometer was used to synchronize the imaging acquisitions with behavior.

## Two-photon imaging during virtual navigation

Two-photon calcium imaging was performed using an Ultima Investigator or an Ultima II scanheads (Bruker, Milan, Italy) equipped with raster scanning galvanometers (6 mm or 3 mm), a 16x/0.8 NA objective (Nikon, Milan, Italy) and multialkali photomultiplier tubes. For GCaMP6f imaging, the excitation source was a Chameleon Ultra pulsed laser tuned at 920 nm (80 MHz repetition rate, Coherent, Milan, Italy). Simultaneous GCaMP6f and jRCaMP1a imaging was performed with a two-beam path configuration in which two laser beams of

different wavelength simultaneously illuminated the sample. On the Ultima Investigator, two pulsed laser sources were combined through a dichroic mirror (zt98rdc-UF1, Chroma Technology, Bellow Falls, Vermont, USA; $\lambda_1$ = 920 nm, Alcor 920 fiber laser—80 MHz repetition rate, Spark Lasers, Martillac, France; $\lambda_2$ = 1060 nm, Chameleon Ultra II—80 MHz repetition rate, Coherent). On the Ultima II, two orthogonally polarized pulsed laser sources were combined through a polarizing beam splitter (05FC16PB.5, Newport; $\lambda_1$ = 920 nm, Chameleon Ultra II; $\lambda_2$ = 1,100 nm, Chameleon Discovery—80 MHz repetition rate, Coherent). Laser beam intensity was adjusted using Pockel cells (Conoptics, Danbury, Connecticut, USA). Imaging average power at the objective outlet was approximately 80 to 110 mW. Fluorescence emission was collected by multialkali PMT detectors downstream of appropriate emission filters (525/70 nm for GCaMP6f, 595/50 nm for jRCaMP1a). Detector signals were digitalized at 12 bits. Imaging sessions were conducted in raster scanning mode at approximately 3 Hz using 5× optical zooming factor. Images contained 256 pixels × 256 pixels field of view (pixel dwell-time, 4 μs; Investigator: pixel size, 0.63 μm; Ultima II: pixel size, 0.51 μm).

One or 2 weeks after surgery, the animals were set on a water restricted schedule, receiving approximately 1 ml of water per day. Weight was monitored daily and remained between 80 and 90% of the starting weight throughout all procedures. Mouse habituation to the experimenter (handling) started 2 days after water scheduling and lasted for a minimum of 2 sessions. Following handling, mice were habituated to the virtual reality setup in successive training sessions. Starting from the second habituation session, the animals were head-tethered for a progressively increasing amount of time, reaching 1 hour in approximately 1 week. During virtual reality training sessions, mice were exposed to the noise generated by the two-photon imaging setup (e.g., galvanometer scanning noise and shutter noise). Training in the virtual environment lasted until animals routinely ran along the linear track. On experimental days, mice were head-tethered, and the virtual reality session started after a suitable field of view was identified. Three to six temporal series (t-series; 750 frames/series; t-series duration, approximately 250 seconds), interleaved by 5 minutes breaks, were acquired during an approximately 1-hour virtual navigation session. Astrocytes and neurons were simultaneously recorded form the same focal plane. At the end of each imaging session, animals were returned to their home cage.

## Data analysis

**Motion correction, image segmentation, and trace extraction.** Analysis was performed using Python 3.6 (python.org) and custom code. t-series were preprocessed to correct motion artifacts using an open-source implementation of up-sampled phase cross-correlation [62,63]. Each t-series was motion corrected using its median projection as reference frame. Corrected t-series were then concatenated in a single movie and, to compensate small x-y drifts across t-series, motion corrected using its median projection image as the reference frame. Regions (typically at the edges of the field of view) within which artifacts could not be corrected were not considered for analysis.

For astrocytic recordings, ROI segmentation was performed on median projections after motion correction using manual annotation. Astrocytic ROIs were classified as soma or process according to visible anatomic features. For each ROI, fluorescence signals were computed as

$$\frac{\Delta F}{F_0} = \frac{F(t) - F_0(t)}{F_0(t)}, \tag{Eq 1}$$

where F(t) is the average fluorescence signal of a given ROI at time t, and $F_0(t)$ is the baseline

fluorescence, computed as the 20th percentile of the average fluorescence intensity in a 30 second-long rolling window centered in t.

For neuronal recordings, cell identification was performed on the median temporal projection of each t-series, after motion correction, by identifying rectangular boxes containing the neuronal soma of the identified neuron, as in [64]. Within the rectangular box, pixels were ranked according to the pixel signal-to-noise (SNR) using the following formula:

$$SNR_{i,j} = \frac{max_{i,j} F_{i,j}(t)}{noise_{i,j}}, \tag{Eq 2}$$

where max $F_{i,j}(t)$ is the maximum fluorescence intensity of the pixel $i,j$, at time $t$, and $noise_{i,j}$ was computed as the standard deviation across all fluorescence values of the t-series below the 25th percentile of the fluorescence distribution of the pixel $i,j$ [64]. Only pixels with SNR value greater than the 80th percentile of the SNR distribution were considered as part of the ROI corresponding to the considered rectangular box. The neuropil signal was computed as the average trace of all pixels in the time series not belonging to bounding boxes. This value was multiplied by a factor r = 0.7 [18] and then subtracted from each fluorescence trace. $\Delta F/F_0$ traces were computed as

$$\frac{\Delta F}{F_0} = \frac{F(t) - F_0(t)}{F_0(t)}, \tag{Eq 3}$$

where F(t) is the neuropil-subtracted fluorescence trace signal at time t, and $F_0(t)$ is the baseline trace computed as 20th percentile of the average intensities in a 10-second rolling window centered in t.

**Identification of calcium events.** For both astrocytic and neuronal fluorescence traces, extraction of statistically significant calcium events was performed on $\Delta F/F_0$ traces via modified implementation of the algorithm described in [25]. In brief, for each trace, a first parameter ($\sigma_1$) was computed as the standard deviation of the whole signal. Values crossing the threshold set at $\pm \sigma_1$ were removed from the trace and a second parameter ($\sigma_2$) was computed as the standard deviation of the thresholded trace. This procedure avoided biases induced by large signal transients on the estimation of the signal standard deviation in the absence of transients and provided a better estimation of signal baseline fluctuations ($\sigma_2$). For astrocytic traces, fluorescence transients were identified on the original trace (thus considering all data) as events that (i) crossed the threshold of $\pm 2\sigma_2$; and (ii) returned within $\pm \sigma_2$ in more than 0.5 s. For neuronal traces, fluorescence transients were identified on the original trace as events that: $i$) crossed the threshold of $\pm 3\sigma_2$; $ii$) returned within $\pm 2\sigma_2$ in more than 0.5 s [25]. For both astrocytic and neuronal signals, these criteria were selected to obtain a false discovery rate <5%, according to the following:

$$FDR = \frac{N_{E_n}}{N_{E_p} + N_{E_n}}, \tag{Eq 4}$$

where $N_{E_p}$ and $N_{E_n}$ are the numbers of identified positive and negative deflections of the $\Delta F/F_0$ trace, respectively. For all subsequent analysis, an event trace was obtained from the $\Delta F/F_0$ trace by setting all fluorescence values outside of those belonging to positive events to 0.

**Identification of reliable spatial modulation of calcium signals.** To evaluate if and how position in the virtual corridor modulated calcium signals, we applied 2 basic requirements: that activity carried significant information about position and that the spatial modulation properties were reliably reproducible across subsets of trials. We restricted the analysis to

running trials, defined as consecutive frames of forward locomotion in which mouse speed was greater than 1 cm/s. Running trials separated by less than 1 second were merged. The average number of running trials per experimental session was 32 ± 3 trials/session ($N = 18$ experimental sessions) for monodirectional experiments and 30 ± 3 trials/session ($N = 18$ experimental sessions) for bidirectional experiments. Monodirectional running trials were on average longer than bidirectional ones (mean running trial length for monodirectional virtual navigation, 140 ± 3 cm, $N = 18$; mean running trial length for bidirectional virtual navigation: 49 ± 4 cm, $N = 18$). Calcium responses were considered with reliable spatial information if they matched both of the following criteria: (i) response field reliability was greater than 0 (see Spatial reliability of calcium responses); and (ii) mutual information between position and calcium event trace was significant (see Spatial information in calcium signals). The same criteria were applied to astrocytic ROIs and neuronal ROIs.

**Analysis of calcium responses during virtual navigation.** Analysis was performed on all running trials, binning the length of the virtual corridor (number of spatial bins, 80; bin width, 2.25 cm). For each ROI, the occupancy map was built by computing the total amount of time spent in each spatial bin. The activity map was computed as the average fluorescence value in each spatial bin. Both the activity map and the occupancy map were normalized to sum 1 and convolved with a Gaussian kernel (width of the Gaussian, $\sigma$, was equal to 3 spatial bins, which corresponded to 6.75 cm). The response profile of an ROI, $RP$, was defined as the ratio of the activity map over the occupancy map for that ROI. For each $RP$, we identified a response field, $RF$, as follows: (i) the array of local maxima greater than the 25th percentile of the response profile values was selected, $C = (c_0, c_1, \ldots, c_n)$; (ii) the elements of $C$ were used to initialize the fitting of the sum of a set of $n$ parametrized Gaussian functions, with mean at one of the elements of $C$, amplitude ($a$) at $0 \leq a \leq 1$, and standard deviation ($\sigma$) at $0 \leq \sigma \leq 90$ cm; (iii) this set of Gaussian functions was fitted to the response profile to solve a nonlinear least squares problem (curve_fit function from [65]); and (iv) the response field was defined as the Gaussian with the highest amplitude and the response field width was defined as $2\sigma_i$.

Thus,

$$RP \cong \sum_{c_i \in C} a_i e^{-\frac{(x-c_i)^2}{2\sigma_i^2}} \; with \; \begin{cases} 0 \leq c_i \leq 180 \; cm \; \forall \; c_i \in C \\ 0 \leq a_i \leq 1 \; \forall \; a_i \in A \\ 0 \leq \sigma_i \leq 90 \; cm \; \forall \; \sigma_i \in S \end{cases} \quad \text{(Eq 5)}$$

$$RF = a_i e^{-\frac{(x-c_i)^2}{2\sigma_i^2}} \; with \; i = \mathrm{argmax}(A) \quad \text{(Eq 6)}$$

**Reliability and stability of calcium spatial responses.** To quantify spatial reliability of response fields, we computed response profiles subsampling either odd or even running trials. For either fraction of running trials, we estimated response field center ($c_{odd}$, $c_{even}$) and response field half-width ($\sigma_{odd}$, $\sigma_{even}$). We quantified spatial reliability of calcium responses as a similarity index, where the absolute difference of response field centers, obtained with either fractions of the running trials, was inversely weighted by the most conservative estimate of response field width:

$$Reliability = 1 - \frac{|c_{odd} - c_{even}|}{2 \times min(\sigma_{odd}, \sigma_{even})} \quad \text{(Eq 7)}$$

ROIs with reliability greater than 0 were considered reliable (S2 Fig).

To classify response field stability, we computed response profiles subsampling running trials recorded either in the first ($h_1$) or the second ($h_2$) half of the experimental session. For either half of the trials, we computed response field center ($c_{h1}$, $c_{h2}$). ROIs with an absolute difference in response field centers smaller than 15 cm were considered stable [24].

**Spatial information in calcium signals.** We used information theory to quantify our information gain (or reduction of uncertainty) about position obtained by knowing the calcium response [27,66]. We computed the mutual information, *I(S;R)*, between position in the linear track, stimulus (*S*), and the calcium event trace, response (*R*), as follows:

$$I(S; R) = \sum_{s \in S, r \in R} p(r)p(r|s) \; log_2 \frac{p(r|s)}{p(s)}, \tag{Eq 8}$$

with *S* and *R* representing the arrays of all possible discrete values of stimulus or response, *p(s)* the probability of the stimulus *s*, *p(r)* the probability of the response r across all trials to any stimulus, and *p(r|s)* the conditional probability of the responses *r* given presentation of stimulus *s*.

We characterized the effects of discretization on the estimates of mutual information, computing mutual information while changing the number of discrete states (N) for both *S* ($N_S$ = 4, 8, 12, 16, 20, 24, 40, 60, 80, 100, and 160) and *R* ($N_R$ = 2, 3, 4, 5, 8, 10, and 20). For the stimulus, we used a uniform count binning procedure and for the response we used equally spaced bins. Statistical significance of mutual information was tested using a nonparametric permutation test. We randomly permuted the calcium event trace $10^4$ times, removing any relationship between *R* and *S*. We used shuffled traces to compute a null distribution of mutual information values. A mutual information value was considered significant if greater than the 95th percentile of the null distribution. Mutual information values were conservatively corrected for limited-sampling bias subtracting the mean value of the null distribution [67,68]. The results of this analysis for astrocytic ROIs are reported in S2 Fig. To allow robust estimates of mutual information values while preserving adequately fine discretization of position, we used $N_s$ = 12 throughout the manuscript. For single ROIs analysis reported in Figs 1, 2, and 5, we used $N_R$ = 4 to discretize astrocytic calcium event traces and $N_R$ = 2 for binarized neuronal event traces (setting to 1 all the nonzero values as in [69]).

**Response profile standard error.** A Jackknife estimator [70,71] of the astrocytic response profile—$RP_{(.)}$—was computed as the average of n-Jackknife samples obtained by iteratively omitting one running trial from the computation. We used this deterministic approach to compute Jackknife standard error (SE) as a function of ROIs spatial modulation (S3 Fig).

$$RP_{(.)} = \frac{1}{n} \sum_{i=1}^{n} RP_i \; with \; i = 1, \ldots, n \tag{Eq 9}$$

and

$$SE = \sqrt{\frac{n-1}{n} \sum_{i=1}^{n} \left( RP_i - RP_{(.)} \right)^2} \tag{Eq 10}$$

For each ROI, we measured the difference between the response field center computed using the Jackknife estimator of the *RP* and the response profiles computed using either odd or even running trials (Fig 1I).

**Spatial precision of calcium responses.** During monodirectional virtual navigation, we measured the spatial precision of calcium responses (SP) with the method reported in [24]. For each ROI reliably encoding spatial information, we binned the length of the virtual corridor in *m* bins (*m* = 40; bin width, 4.5 cm), and for each running trial (*n*), we calculated the

center of mass ($COM_n$) of the calcium response (Eq 11), where $DF_i$ is the value of the event trace observed in the $i$-th bin and $x_i$ is the center of the $i$-th bin. For each ROI, we then computed the average center of mass across N trials ($COM_w$, Eq 12), weighting each $COM_n$ by the peak amplitude of the event trace during the $n$-th running trial ($A_n$). Spatial precision was computed as the inverse of the trial-by-trial squared difference between $COM_n$ and $COM_w$ weighed by peak amplitude (Eq 13).

$$COM_n = \frac{\sum_m^i DF_i x_i}{\sum_m^i DF_i} \tag{Eq 11}$$

$$COM_w = \frac{\sum_N^n COM_n A_n}{\sum_N^n COM_n} \tag{Eq 12}$$

$$SP = \left( \sqrt{\frac{\sum_N^n A_n (COM_n - COM_w)^2}{\sum_N^n A_n}} \right)^{-1} \tag{Eq 13}$$

When comparing spatial precision of astrocytic and neuronal responses (S16 Fig), we corrected for the different dynamic range (DR) of the 2 genetically encoded calcium indicators. For each imaging session, we estimated $DR_A$ and $DR_N$ as the mean DR for astrocytic and neuronal event traces, respectively (mean ± SEM; $DR_A = 0.95 \pm 0.13$ for 76 astrocytic ROIs expressing GCaMP6f; $DR_N = 0.54 \pm 0.06$ for 335 neuronal ROIs expressing jRCaMP1a; data from 11 imaging sessions from 7 animals). We corrected spatial precision of neuronal responses in each imaging session by the factor $DR_A / DR_N$.

**Directionality of astrocytic spatial responses.** In experiments where the mouse performed bidirectional navigation, astrocytic ROIs could be spatially modulated in either running direction. To quantify whether responses were direction selective, we computed the directionality index (DI) as

$$DI = \frac{\bar{F}_d - \bar{F}_o}{\bar{F}_d + \bar{F}_o}, \tag{Eq 14}$$

where $\bar{F}_d$ was the average of $\Delta F/F_0$ inside the response field, and $\bar{F}_o$ was the average of $\Delta F/F_0$ at the same response field while running in the opposite direction. $DI > 0$ indicated that average response at the response field was direction selective. We compared the distribution of $DI$ values for all spatially modulated ROIs with surrogate data. To this end, we randomly selected one of the informative ROIs and computed $DI$ after applying a random shift of response field position along the linear track while preserving its width. We repeated this operation $10^5$ times, obtaining a distribution of $DI$ values representing the occurrence of $DI$ values at any spatial location as wide as a response field.

**Population analysis using mutual information.** For experiments in which we simultaneously recorded astrocytic and neuronal calcium activity, we used all running trials to compute the mutual information about animals' position obtained by observing the calcium signals of a pair of simultaneously recorded ROIs. Results are reported as a function of pair composition, with pairs containing either two astrocytic ROIs, two neuronal ROIs, or one element of each type.

Mutual information between the spatial position, $S$, and the array of joint responses for a pair of ROIs, $R = (R_1, R_2)$, was computed as [32]

$$I(S; R) = \sum_{s \in S, r \in R} p(\boldsymbol{r}_1, \boldsymbol{r}_2) p(\boldsymbol{r}_1, \boldsymbol{r}_2 | s) log_2 \frac{p(\boldsymbol{r}_1, \boldsymbol{r}_2 | s)}{p(s)}, \qquad \text{(Eq 15)}$$

where $p(s)$ is the probability stimulus $s$, $p(r_1, r_2)$ is the probability of joint responses $r_1$ and $r_2$ across all trials to any stimulus, and $p(r_1, r_2 | s)$ is the conditional probability of the joint responses $r_1$ and $r_2$ given presentation of stimulus $s$.

For consistency with single ROI analysis, spatial position was discretized with $N_s = 12$. To allow consistent scaling of probability spaces and comparable information values, the astrocytic calcium event trace was binned with $N_R = 2$ (we verified that the main conclusions were maintained when using $N_R = 3$ and $N_R = 4$), and $N_R = 2$ for neuronal calcium event trace discretization, as described for single neuron analysis. To correct mutual information bias caused by limited sampling of astrocytic or neuronal responses, we performed quadratic extrapolation correction [31,72] using 100 iterations.

To quantify whether the within-trial correlations of a given ROI pair enhanced the amount of position information carried by the pair, we used trial shuffling to disrupt the within-trial correlations between ROIs while keeping intact the spatial position information of individual ROIs. Within subsets of trials with the same position bin, we generated pseudo-population responses by independently combining shuffled identities of trials for each ROI. Thus, responses of individual ROIs to the spatial position were maintained while within-trial correlations between ROIs were disrupted. We computed 100 trial shuffling estimates of mutual information, $I(S;R)_{\text{trial-shuffled}}$, for calcium responses at fixed position. A pair was classified as having information enhanced by correlations, if $I(S;R)$ was greater than the 95th percentile of the corresponding $I(S;R)_{\text{trial-shuffled}}$ distribution.

**Information breakdown.**   We performed information breakdown analysis [31,32]. We decomposed spatial information carried by a pair of ROIs, $I(S;R)$, into four terms. Each term expressed a different contribution carried by correlations to the information between the ROIs. The decomposition is as follows:

$$I(S; R) = I_{LIN} + I_{SS} + I_{CI} + I_{CD} \qquad \text{(Eq 16)}$$

$I_{LIN}$, the mutual information linear term, is the sum of the information provided by each ROI. $I_{SS}$ (signal similarity term) is a nonpositive term quantifying the decrease of information (amount of redundancy) due to signals correlation caused by correlations between the trial averaged spatial position profiles of the calcium signals of the two ROIs. $I_{CI}$ (stimulus independent correlation) is a term that can be either positive, null, or negative and that quantifies the contribution of stimulus-independent correlations. $I_{CI}$ is negative if noise and signal correlations have the same signs and positive otherwise. $I_{CD}$ (stimulus-dependent correlational term) is a nonnegative term that quantifies the amount of information, above and beyond that carried by the responses of individual ROIs carried by stimulus modulation of noise correlation strength. Although $I_{CD}$ is strictly nonnegative, $I_{CD}$ values could occasionally become slightly negative due to quadratic extrapolation bias correction.

The above calculations of $I(S;R)$ were conducted with a bias correction procedure that, with the typical number of trials per spatial location represented in our data (mean ± SEM 72 ± 7 trials/location), was shown to be accurate for removing the limited sampling bias [73]. However, it was also shown to leave on average, a small residual positive overestimation that tended to slightly overestimate synergy [73]. To make sure that our results of prevalent synergy could not be explained by a residual positive bias, we repeated the calculation with the bias

correction procedure described and termed "shuffled" in [73]. The shuffling correction has a higher variance but overcorrects the bias and leaves overall a smaller residual underestimation of I(S;R). We found that this alternative bias correction procedure generated results similar to the ones presented in the paper (S19 Fig, S3 Table). The fact that our findings are stable using two methods biased in opposite directions shows that our information estimations are accurate and that the results are solid and conservative.

**Position-dependent correlation.** To measure whether correlation between pairs of neuronal and astrocytic ROIs was position-dependent, we computed pairwise Pearson correlations between calcium signals sampled inside and outside the response fields. On average, response fields were smaller than half the linear track, thus either set of observations, inside or outside the response field, could contain uneven amounts of datapoints. To compensate for the unbalanced numerosity, we resampled the same number of points found in the smaller set, while preserving temporal ordering. We then computed Pearson correlation between the two vectors. For each pair of ROIs, we computed the average Pearson correlation with 100 iterations of this procedure. We repeated this procedure inside both astrocytic fields and neuronal response fields.

**Population analysis using SVM decoder of spatial position.** To decode animals' position from a population of ROIs, we trained an SVM classifier with Gaussian kernel [74–76]. We performed decoding analysis on three datasets: (i) astrocytic signals during monodirectional virtual navigation; (ii) astrocytic signals during bidirectional virtual navigation; and (iii) simultaneous recording of astrocytic and neuronal signals during monodirectional virtual navigation. Experimental sessions were considered independently. We evaluated decoding performance as a function of decoding granularity, $G$, i.e., the number of spatial bins we used to discretize the linear track. For monodirectional virtual navigation, we used G = (4, 8, 12, 16, 20, 24), and for bidirectional virtual navigation, for which there was a limited number of running trials, we used G = (4, 8, 12, 16). All experimental sessions with at least 3 observations in each spatial bin were included in the analysis. For experiments in which we recorded astrocytic and neuronal calcium activity simultaneously, we measured decoding performance for multiple population settings, using both astrocytic and neuronal signals, or excluding either one.

We used experimental session as the n-dimensional array of calcium event traces ($N$ = number of ROIs) to decode discretized positions along the virtual linear track at each time point. Each experimental session was composed of a set of $T_{exp}$ observations $(X_i, y_i)$, where $X_i$ is the n-dimensional array of the calcium activity of the n ROIs, whereas $y_i$ corresponds to the discretized spatial position. For each granularity, we trained and tested the SVM using 10-fold cross-validation procedure on each experimental session independently. During each iteration of the cross-validation, the SVM was trained and optimal hyperparameters were selected performing 5-fold cross-validation on each fold training set. Predictions of the decoder for each of the 10-folds used as test were then collected to compute the overall performance of the decoder.

For each granularity, we measured decoding performance computing decoded information, as the mutual information between predicted and real spatial position [27]:

$$I(S; S_p) = \sum_{s,s_p} p(s; s_p) \, log_2 \frac{p(s; s_p)}{p(s)p(s_p)}, \tag{Eq 17}$$

where $s_p$ denotes the decoded spatial position (with the SVM method described above) from the population response vector in each trial, $s$ is the actual spatial position of the animal, and $p$ $(s; s_p)$ is the decoder's confusion matrix obtained from the predictions of the 10-folds cross-

validation test set. We corrected mutual information measures for the limited sampling bias using the conservative bootstrap correction method described in [67,68,73].

Decoding performance was also computed as decoding accuracy (fraction of correct predictions):

$$Accuracy = \frac{number\ of\ correct\ predictions}{total\ number\ of\ predictions} \tag{Eq 18}$$

To assess the statistical significance of decoding results, we trained and tested the decoder on each experimental session after randomly permuting position and responses. This procedure removed all information about position carried in the responses. We performed $10^3$ random permutations for each granularity and population type. We then used the distribution of information values on permuted data as the null hypothesis distribution for the one-tailed nonparametric permutation test of whether information was significantly larger than zero. We repeated this procedure separately for each granularity.

To assess if the correlations among neurons and/or astrocytes increased the amount of spatial information, we disrupted across neuron correlations by randomly shuffling, separately for each ROI, the order of trials with the same position category. We performed 500 trial shuffling for each granularity and population type. We then used the trial-shuffled distribution as the null hypothesis distribution for the one-tailed nonparametric permutation test of whether the information in the real population vector (which includes correlations) is significantly higher than that obtained when correlations are removed.

**Decoding error analysis.** We investigated classification errors made by the decoder for each decoding granularity. We considered only misclassified samples in the test set, and we measured the distance between the position predicted by the decoder and the ground truth position. We computed the frequency histogram of these deviations from the ground truth, and fitted a Gaussian curve [65] using nonlinear least squares. For each histogram, we computed $R^2$ score to quantify the fitting performance.

**Computing genuine spatial information that cannot be possibly attributed to visual cue information.** To assess whether spatial information encoded in a calcium response could be attributed to visual cues, we leveraged on the structure of the visual patterns of the virtual linear track. Three distinct visual cues covered the whole length of the corridor each in 60 cm–long segments. Within each segment, the visual stimuli were periodically repeated (Fig 1C). We reasoned that, if responses in the virtual reality corridor were only modulated by visual cues, regardless of the position in which the visual stimulus was provided, then it would not be possible to discriminate between positions within the spatial extent of each visual cue (60 cm). In such case, the responses would not carry any spatial information above and beyond the one that is inherited from the information they carry about the identity of the visual cue. To quantitatively test this hypothesis, we computed mutual information using (Eq 8), while randomly shuffling positions of calcium responses observed within the spatial extent of each visual cue ($I_V$). This spatially targeted permutation procedure preserved the information about visual cues identity, while it destroyed all the genuine spatial information carried by the response beyond visual cue information. We repeated this spatially targeted permutation procedure generating a distribution of $I_V$ values for each ROI (100 permutations). Information values were corrected for the limited sampling bias using the Panzeri–Treves procedure [68]. Responses were considered as carrying information beyond visual cue identity if the real information, I, was greater than 95th percentile of the distribution of $I_V$. Positions and responses were discretized using uniform width bins. We systematically characterized the effect of position discretization on the estimates of I and $I_V$ repeating binning spatial positions into

different number of spatial bins $N_S$ ($N_S$ = 9, 12, 15 18). We used numbers of bins that were multiple integers of 3 to ensure that each spatial bin fell within an individual 60 cm–long visual cue zone. The number of response bins $N_R$ for astrocytic and neuronal responses were 4 and 2, respectively.

We extended this analysis measuring information on population response vectors. We trained and tested the SVM decoder to decode discretized positions along the virtual linear track from the n-dimensional array of calcium event traces on each experimental session, while performing the spatially targeted permutation procedure described above. We repeated the permutation procedure 500 times to build a distribution of decoded information (Eq 17) to estimate $I_V$. For each experimental session, we computed the mean value of decoded information as the average of $I_V$ distribution. We repeated this procedure systematically varying the value of decoding granularity G (again using multiple integers of 3 for the number of spatial bins, thus leading to using G = 9, 12, 15, 18).

**Histology.** Histology preparations were obtained similarly to [47,77]. In brief, animals were deeply anesthetized with urethane and transcardially perfused with 0.01 M phosphate buffered saline (PBS; pH 7.4) and then 4% paraformaldehyde (PFA) in phosphate buffer (PB; pH 7.4). Brains were postfixed overnight (ON) at 4°C and subsequently cut to obtain coronal slices of 40 to 50 μm thickness. Sections were incubated ON, or for 48 hours, at 4°C in primary antibody diluted in a PBS solution containing 5% normal goat serum, 0.3% Triton X-100. Sections were then incubated for 24 hours at 4°C in the appropriate secondary antibody solution. Cell nuclei were counterstained incubating the sections with Hoechst 33342 (1: 300) for 20 minutes at room temperature, mounted on glass slides using Fluoromount (Sigma-Aldrich, Saint Louis, Missouri, USA) and coverslipped. Primary antibodies were Anti-GFAP (1:300 rabbit, Abcam Ab16997, Cambridge, United Kingdom), Anti-NeuN (1:250 mouse, Millipore MAB377, Billerica, Massachusetts, USA), Anti-GFP (1:500, chicken, Abcam Ab13970). Alexa-conjugated (Invitrogen, Carlsbad, California, USA) secondary antibodies were used.

Fluorescence images were acquired with either a Leica SP5 inverted confocal microscope (40x/1.25 NA immersion objective, Leica, Milan, Italy) or with a Nikon A1 inverted confocal microscope (20x/0.8 NA objective, Nikon). Hippocampal regions and layers were identified using the anatomical hallmarks provided by cell nuclei counterstaining (S1A, S1B, S11A and S11B Figs).

To quantify the extent of astrocytic reactivity in the hippocampus of implanted animals, we bilaterally acquired image series of the hippocampal formations (3 × 3 tiles, 1024 × 1024 pixels/tile, 154 pixels overlap, pixel size 0.62 μm/pixel, 8 planes, 1.5 μm/step; S1A and S1B Fig top). To avoid biases, image series of both hemispheres (the implanted one and the control one) were acquired with the same parameters (e.g., excitation laser power and photodetectors gain). We estimated the fraction of tissue immunoreactive for GFAP on maximum intensity projections. For each pair of projections (one for the implanted hemisphere and one for the nonimplanted one), we selected 3 similar ROIs extending along the mediolateral axis of the hippocampal formation and spanning the dorsoventral extent of either stratum Oriens, stratum Pyramidale, or stratum Radiatum (S1A–S1D Fig). ROIs selected on each hemisphere were identical. We performed image thresholding on pairs of ROIs (one from the implanted and one from the control hemisphere) from matching hippocampal strata, selecting as cutoff value the maximum of the threshold values computed on either ROI with the triangle method [78]. Thresholded ROIs were used to compute the fraction of GFAP immunolabeled pixels and their average fluorescence intensity value.

Selectivity of Genetically-encoded calcium indicators (GECI) expression was assessed on confocal z-image series (9 planes, 2 μm/step) using ImageJ (imagej.nih.gov/ij [79]) and the CellCounter plugin, counting cells immunolabeled for either GFAP or NeuN among GECI-expressing cells.

**Statistics.** Significance threshold for statistical testing was always set at 0.05. No statistical methods were used to predetermine sample size, but sample size was chosen based on previous studies [17,23,24]. Statistical analysis was performed using Python (SciPy 0.24, NumPy 1.19, statsmodels 0.9) or the InfoToolbox library [31] available for Matlab (MathWorks R2019b). A Python 3 [80] (version 3.6) front-end was used for execution. To test for normality, either a Shapiro–Wilks (for $N \leq 30$) or a D'Agostino K-squared test (for $N > 30$) was run on each experimental sample. When comparing 2 paired populations of data, a paired $t$ test or Wilcoxon signed rank test were used to calculate statistical significance (for normal and nonnormal distributions, respectively). Independent samples $t$ test and two-sample Kolmogorov–Smirnov test or Wilcoxon rank sums test were used for unpaired comparisons of normally and nonnormally distributed data, respectively. A binomial test was used to test if the fraction of successes at the population level in a number of statistical test performed at $p = 0.05$ could be due to chance. Bonferroni correction was applied to correct for the multiple testing problem when appropriate. Surrogate data testing was performed as described in the specific methods sections. All tests were two-sided, unless otherwise stated. When reporting descriptive statistics of data distributions, we used either the mean ± standard deviation (mean ± SD) for normal data or the median ± median absolute deviation (median ± MAD) for nonnormal data. Datasets reporting average values across experimental sessions were presented as mean ± standard error of the mean (mean ± SEM). Bootstrap estimation was performed to identify 95% confidence intervals for mean values and for mean differences [81], where appropriate [82]. Effect size was quantified as Cohen d coefficient [83].

## Supporting information

**S1 Fig. Chronic CA1 window to monitor astrocytic calcium dynamics in head restrained mice. (A, B)** Representative images of hippocampal brain slices from animals injected with AAV5 pZac2.1 gfaABC1D-cyto-GCaMP6f and implanted with a chronic optical window. Images are maximum intensity projection of confocal z-stacks (8 planes, 1.5 μm/step) from hemispheres contralateral (A) and ipsilateral (B) to the injection and implant site. Brain slices were stained with anti-GFAP and anti-GFP primary antibodies, which were counterstained with Alexa-546 and Alexa-488 conjugated secondary antibodies, respectively. Cell nuclei were labeled with Hoechst. **(C, D)** Zoom-in of the ROIs (white rectangles in A and B) used for quantification of GFAP-staining in stratum Oriens, stratum Pyramidale, and stratum Radiatum. **(E)** Fraction of ROI area immunolabeled for GFAP. **(F)** Average fluorescence intensity of GFAP-positive pixels in the 3 hippocampal regions under the different experimental conditions. Data are presented as mean ± SD from 13 slices in 3 animals. In E: $p = 1.4E-2$, $p = 1.3E-1$, and $p = 8.2E-1$ for stratum Oriens, Pyramidale, and Radiatum, respectively. Paired $t$ test. In F: $p = 8.8E-2$, $p = 9.5E-1$, and $p = 2.0E-1$ for stratum Oriens, Pyramidale, and Radiatum, respectively. Paired $t$ test. **(G)** Fraction of GCaMP6f cells immunolabeled for GFAP (95 ± 7%, out of a total of 45 GCaMP6f-expressing cells from $N = 6$ sections from 3 mice). Scale bars: 200 μm and 50 μm for A and B and C and D, respectively. The data presented in this figure can be found in S2 Data. GFAP, glial fibrillary acidic protein; GFP, green fluorescent protein; ROI, region of interest.
(TIFF)

**S2 Fig. Identification of reliable spatial modulation of astrocytic calcium signals. (A)** Minimum response field width between even and odd trials as a function of the difference in place field position. The pseudocolor scale indicates reliability of the response (see Methods). **(B, C)** Mutual information values (B) and fraction of ROIs showing significant spatial information (C) as a function of the number of bins for the stimulus (animals' position in the linear track).

Colors indicate different binning of the response (calcium event trace). Mutual information values were bias-corrected using bootstrap method ($10^4$ iterations). Significance level for information content was set at $p < 0.05$. **(D)** Fraction of ROIs with reliable spatial information as a function of the number of bins for the stimulus. Colors indicate different binning of the response. Data in (B, D) are presented as mean ± SEM from 7 imaging sessions in 3 animals. The data presented in this figure can be found in S2 Data. ROI, region of interest; SEM, standard error of the mean.
(TIFF)

**S3 Fig. Reliable spatial modulation of astrocytic calcium signals. (A)** Representative traces showing calcium signals for 5 astrocytic ROIs encoding spatial information shown in Fig 1E. Top: Solid black lines indicate the average astrocytic calcium response across runs as a function of spatial position, and the dashed gray lines indicate response field Gaussian fitting function. Bottom: Solid gray lines indicate normalized calcium event traces as a function of position in the virtual corridor for individual runs. Filled gray areas indicate response field width. **(B)** Cumulative distribution of the mean SE of the response profile in astrocytic ROIs (median ± MAD 1.3E-2 ± 1.2E-2 cm$^{-1}$, $N = 155$ out of 356 total ROIs, for ROIs with reliable spatial information, black; 1.8E-2 ± 2.0E-2 cm$^{-1}$, $N = 201$ out of 356 total ROIs, for not modulated ROIs, gray: $p = 1E-5$, Kolmogorov–Smirnov test). **(C)** Cumulative distribution of Pearson correlation values between astrocytic response profiles in even and odd trials (median ± MAD 0.63 ± 0.24, $N = 155$ out of 356 total ROIs for ROIs with reliable spatial information, black; 0.19 ± 0.37, $N = 201$ out of 356 total ROIs, for not modulated ROIs, gray; $p = 5E-14$, Kolmogorov–Smirnov test). **(D)** Cumulative distribution of the spatial precision index of the response field of astrocytic ROIs (black: median ± MAD 3.2E-2 ± 0.6E-2, $N = 155$ out of 356 total ROIs, for ROIs with reliable spatial information; gray: 3.0E-2 ± 0.5E-2 cm$^{-1}$, $N = 201$ out of 356 total ROIs, for not modulated ROIs: $p = 3.8E-2$, Kolmogorov–Smirnov test). In all panels, data from 7 imaging sessions in 3 animals. The data presented in this figure can be found in S2 Data. MAD, median absolute deviation; ROI, region of interest; SE, standard error.
(TIFF)

**S4 Fig. Modulation of astrocytic calcium responses during locomotion and virtual navigation. (A)** Scatterplot of the average $\Delta F/F_0$ of astrocytic ROIs during baseline (mouse speed $\leq 1$ cm/s) versus during locomotion (mouse speed $> 1$cm/s). Under both conditions, the mouse was immersed in the virtual reality. Black open dots represent averages of each imaging session. The red cross shows the mean ± SEM of plotted data (mean $\Delta F/F_0$ during baseline 0.14 ± 0.01; mean $\Delta F/F_0$ during locomotion 0.25 ± 0.03; $N = 356$ ROIs; $p = 0.016$ Wilcoxon signed rank test). **(B)** Same as in (A) but for $\Delta F/F_0$ values measured in astrocytic ROIs encoding reliable spatial information when the mouse was not exposed to the visual stimulation of the virtual reality (during intertrial intervals) versus when the mouse was passing through each ROIs' response fields (mean $\Delta F/F_0$ during without visual stimulation 0.21 ± 0.03; mean $\Delta F/F_0$ inside the response field 0.37 ± 0.04; $N = 155$ out of 356 total ROIs; $p = 0.016$ Wilcoxon signed rank test). Data in (A, B) from 7 imaging sessions in 3 animals. The data presented in this figure can be found in S2 Data. ROI, region of interest; SEM, standard error of the mean.
(TIFF)

**S5 Fig. Calcium signals of CA1 astrocytes encode direction selective spatial information during virtual bidirectional navigation. (A)** Two-photon functional imaging of CA1 astrocytes is performed during bidirectional virtual navigation. **(B)** Head-restrained mice run on an air-suspended spherical treadmill in a linear virtual track in both forward and backward

directions. Water rewards are delivered at either end of the virtual corridor. **(C)** Median projection of GCaMP6f-labeled astrocytes in the CA1 pyramidal layer. White lines indicate segmented ROIs. Scale bar: 20 μm. **(D)** Calcium signals for 5 representative astrocytic ROIs reliably encoding spatial information across the corridor length. Solid black lines indicate the average astrocytic calcium response across trials as a function of spatial position. Dashed gray lines and filled gray areas indicate the Gaussian fitting function and the response field width (see Methods), respectively. **(E)** Normalized astrocytic calcium responses as a function of position for astrocytic ROIs with reliable spatial information. Trials are divided according to running direction (forward and backward). For forward trials, informative ROIs are $N = 192$ out of 648 total ROIs, mean ± SD: 29 ± 13%; for backward trials, informative ROIs are $N = 133$ out of 648 ROIs, mean ± SD: 20 ± 13%, $p = 0.09$, Wilcoxon signed rank test. Scale bar: 20 ROIs. Yellow dots indicate the center position of the response field, and the magenta dots indicate the width of the field response. **(F)** Distributions of astrocytic response field position for forward and backward running direction. Median ± MAD 93 ± 66 cm, $N = 192$ out of 648 total ROIs for the forward direction; 138 ± 47 cm $N = 133$ out of 648 total ROIs for the backward direction; $p = 9E-7$, Kolmogorov–Smirnov test). **(G)** Distributions of response field width for the forward and backward running direction (response field width, 44 ± 19 cm, $N = 192$ out of 648 total ROIs for the for forward direction; response field width, 44 ± 28 cm, $N = 133$ out of 648 total ROIs for the backward direction; $p = 0.34$, Wilcoxon rank sums test). **(H)** DI for forward and backward running directions (DI, 0.18 ± 0.16, $N = 192$ out of 648 total ROIs for forward trials; DI, 0.16 ± 0.16, $N = 133$ out of 648 total ROIs for backward trials; $p = 8E-19$ and $p = 2E-8$, respectively, Kolmogorov–Smirnov test versus shuffled distribution). In all panels, data from 18 imaging sessions in 4 animals. The data presented in this figure can be found in S2 Data. DI, directionality index; MAD, median absolute deviation; ROI, region of interest; SD, standard deviation; SEM, standard error of the mean.
(TIFF)

**S6 Fig. Anatomical organization of subcellularly localized astrocytic calcium signals. (A)** Distribution of field position for soma ROIs and process ROIs ($p = 0.36$, Kolmogorov–Smirnov test). **(B)** Distribution of response field width for astrocytic soma ROIs and process ROIs (median width for soma ROIs: 60 ± 19 cm; median width for process ROIs: 56 ± 22 cm, $p = 0.36$, Wilcoxon rank sums test). **(C)** For each pair of ROIs within a given astrocyte, the distance (d) between the centers of 2 ROIs and the angle between the line connecting the 2 ROI centers and the x-axis are calculated. Only astrocytes showing significant spatial modulation in the soma and at least 1 process were used for this analysis. **(D, E)** Difference in field position of a process with respect to the field position of its corresponding soma, expressed as function of Cartesian (D) and polar (E) coordinates of the ROI centers. **(F)** Difference in response field position of a process with respect to the field position of its corresponding soma as a function of the process distance from cell soma ($R^2 = 0.01$, $p = 3.3E-1$, Wald test, data from 19 cells from 7 imaging sessions on 3 animals). **(G, H)** Absolute value (G) or signed (H) difference in response field position of a process ROI with respect to the field position of its corresponding soma as a function of the process angular coordinate (absolute value of difference in response field $R^2 = 0.01$, $p = 4.8E-1$, Wald test; signed value of difference in response field $R^2 = 0.01$, $p = 4.1E-1$, Wald test, data from 19 cells from 7 imaging sessions on 3 animals). The data presented in this figure can be found in S2 Data. ROI, region of interest.
(TIFF)

**S7 Fig. Temporal relationships among subcellularly localized astrocytic calcium signals. (A)** Event triggered average of astrocytic calcium responses. Calcium responses of putative receiver (R) ROIs are aligned to calcium events of putative source (S) ROIs according to

anatomic identities of ROIs (e.g., somatic receiver ROIs and somatic source ROIs). Data from 7 imaging sessions in 3 animals. Black line indicates the mean, and shaded area the standard deviation. **(B–D)** Same as in (A) for pairs of process receiver and somatic source (B), somatic receiver and process source (C), and process receiver and process source (D). **(E–G)** Same as in (B-D) but for pairs of ROIs belonging to the same astrocyte ($N$ = 46 astrocytes from 7 imaging sessions in 3 animals). **(H)** Response time (see Methods) for signals shown in (A-D). $p$ = 6E-4, Friedman test with Nemenyi post hoc correction. **(I)** Response time for signals shown in (E-G). $p$ = 7E-3, Friedman test with Nemenyi post hoc correction. The data presented in this figure can be found in S2 Data. ROI, region of interest.
(TIFF)

**S8 Fig. Decoding animal's position from astrocytic calcium signals in the unidirectional virtual navigation task. (A)** Decoding accuracy as a function of spatial granularity on real (white), chance (dark gray), and trial-shuffled (gray) data (see Methods). Data are presented as mean ± SEM from 7 imaging sessions on 3 animals; see also S2 Table. The data presented in this figure can be found in S2 Data. SEM, standard error of the mean.
(TIFF)

**S9 Fig. Decoding animal's position from astrocytic calcium signals in the bidirectional virtual navigation task. (A)** Confusion matrices of an SVM classifier for different spatial granularities (G = 4, 8, 12, 16) for trials in which the mouse was running in the forward direction (forward). The actual position of the animal is shown on the x-axis, the decoded position on the y-axis. Gray scale indicates the number of events in each matrix element. **(B)** Decoded information as a function of spatial granularity on real (white) and chance (gray) data for forward trials. **(C)** Decoding accuracy as a function of spatial granularity. **(D)** Decoding error as a function of the error position within the confusion matrix for forward trials. The color code indicates spatial granularity. In panels (A–D), data from 15 imaging sessions in 4 animals. **(E–H)** Same as in (A–D) for trials in the backward direction. Data from 17 imaging sessions in 4 animals. In (B, C, F, G), data are presented as mean ± SEM. See also S6 Table. The data presented in this figure can be found in S3 Data. SEM, standard error of the mean; SVM, support vector machine.
(TIFF)

**S10 Fig. Visual cues identities do not explain animal's position decoding from astrocytic calcium signals. (A)** Confusion matrices of an SVM classifier decoding the mouse's position using population vectors data comprising astrocytic ROIs in which position was shuffled within visual cues. Shuffling position within visual cues decoupled spatial information encoded in the population vector from the information related to visual cues identity (see Methods). The true position of the animal is shown on the x-axis and the decoded position on the y-axis. Gray scale indicates the percentage of occurrence of each matrix element (Decoding). Results are shown for various spatial granularities (G = 9, 12, 15, 18). In all panels, data from 500 permutations on 7 imaging sessions in 3 animals are shown. The data presented in this figure can be found in S3 Data. ROI, region of interest; SVM, support vector machine.
(TIFF)

**S11 Fig. Chronic CA1 window to simultaneously monitor astrocytic and neuronal calcium signals in head restrained mice. (A, B)** Representative images of hippocampal CA1 areas from animals transduced with AAV5 pZac2.1 gfaABC1D-cyto-GCaMP6f and AAV1/2 pAAV CAMKII-jRCaMP1a implanted with a chronic optical window. Images are maximum intensity projection of confocal z-stacks (9 planes, 2 μm/step) from brain slices stained either with anti-GFAP (A) or an anti-NeuN primary antibody (B). In both cases, counterstaining was

performed with an Alexa-647 conjugated secondary antibody. **(C)** Related to (A): Fraction of GCaMP6f-expressing cells immunolabeled for GFAP (100 ± 0%, out of a total of 71 GCaMP6f-expressing cells from $N$ = 6 sections from 3 mice). **(D)** Related to (B): fraction of jRCaMP1a-expressing cells immunolabeled for NeuN (93 ± 8%, out of a total of 985 jRCaMP1a-expressing cells from $N$ = 6 sections from 3 mice). **(E)** Same as in (D) but for GCaMP6f-expressing cells (0 ± 0%, out of a total of 50 GCaMP6f-expressing cells, from $N$ = 6 sections from 3 mice). Scale bars in A and B: 50 μm. The data presented in this figure can be found in S3 Data. GFAP, glial fibrillary acidic protein.
(TIFF)

**S12 Fig. Temporal relationships between astrocytic and neuronal signals. (A–D)** Event triggered average of astrocytic calcium responses. Calcium responses of putative receiver (R) ROIs are aligned to calcium events of neuronal PCs. Astrocytic receiver ROIs could be in the soma (s) or processes (p). Neuronal cells were classified as being close (≤15 μm) or far (>15 μm) from astrocytic receiver ROIs. Data from 11 imaging sessions in 7 animals. The black line indicates the mean, the shaded area the standard deviation. **(E, F)** Same as in (A–D) but for receiver ROIs belonging to the same astrocyte ($N$ = 23 cells from 11 imaging sessions in 7 animals). **(I–L)** Same as in (A–D) but calcium responses of putative receiver (R) ROIs are aligned to calcium events of nonspatially informative cells (non-PC). Data from 11 imaging sessions in 7 animals. **(M–P)** Same as in (I–L) but for receiver ROIs belonging to the same astrocyte ($N$ = 48 astrocytes from 11 imaging sessions in 7 animals). The data presented in this figure can be found in S3 Data. PC, place cell; ROI, region of interest.
(TIFF)

**S13 Fig. Modulation of astrocytic and neuronal calcium responses during locomotion and virtual navigation. (A)** Scatterplot of the average $\Delta F/F_0$ of astrocytic ROIs during baseline (mouse speed ≤ 1 cm/s) versus during locomotion (mouse speed > 1 cm/s). Under both conditions, the mouse was immersed in the virtual reality. Black open dots show averages of each imaging session. The red cross shows the mean ± SEM of plotted data. Average $\Delta F/F_0$ values were measured in astrocytic ROIs (left; mean $\Delta F/F_0$ during baseline 0.06 ± 0.01; mean $\Delta F/F_0$ during locomotion 0.10 ± 0.01, $N$ = 341 ROIs; $p$ = 9.8E-4 Wilcoxon signed rank test) and neuronal ROIs (right; mean $\Delta F/F_0$ during baseline 0.017 ± 0.003; mean $\Delta F/F_0$ during locomotion 0.03 ± 0.01, $N$ = 870 ROIs; $p$ = 9.8E-4 Wilcoxon signed rank test) recorded from mice co-injected with AAV5 pZac2.1 gfaABC1D-cyto-GCaMP6f and AAV1/2 pAAV-CAMKII-jRCaMP1a. **(B)** Same as in (A) but for $\Delta F/F_0$ values measured in ROIs encoding reliable spatial information when the mouse was not exposed to the visual stimulation of the virtual reality versus when the mouse was passing through each ROIs' response fields. Astrocytic ROIs, left, (mean $\Delta F/F_0$ without visual stimulation 0.07 ± 0.01; mean $\Delta F/F_0$ inside the response field 0.13 ± 0.02; $p$ = 0.016 Wilcoxon signed rank test), neuronal ROIs, right, (mean $\Delta F/F_0$ without visual stimulation 0.020 ± 0.002; mean $\Delta F/F_0$ inside the response field 0.07 ± 0.01; $p$ = 0.016 Wilcoxon signed rank test). Data in (A, B) are presented as mean ± SEM and come from 11 imaging sessions in 7 animals. The data presented in this figure can be found in S3 Data. ROI, region of interest; SEM, standard error of the mean.
(TIFF)

**S14 Fig. Pairwise correlations of calcium signals during virtual navigation. (A, B)** Pearson correlation for different pairs of ROIs. Pairs were composed either of two astrocytic ROIs belonging to the same astrocyte (A-A_same), two astrocytic ROIs belonging to the different astrocytes (A-A_other), two neuronal ROIs (N-N), or one astrocytic and one neuronal ROI (A-N). Red line indicates the zero correlation level. In (A), only results for ROI pairs with

reliable spatial information are reported ($p$ = 5.2E-3, $p$ = 6.5E-4, $p$ = 9.4E-4, $p$ = 1.5E-2, $p$ = 1.5E-2, $p$ = 9.9E-2 for A-A$_{same}$ versus A-A$_{other}$, A-A$_{same}$ versus N-N, A-A$_{same}$ versus A-N, A-A$_{other}$ versus N-N, A-A$_{other}$ versus A-N, N-N versus A-N, respectively. Wilcoxon rank sums test with Bonferroni post hoc correction). In (B), results for all possible pairs are displayed ($p$ = 2.6E-3, $p$ = 4.3E-4, $p$ = 4.3E-4, $p$ = 8.6E-4, $p$ = 8.6E-4, $p$ = 1.4E-1 for A-A$_{same}$ versus A-A$_{other}$, A-A$_{same}$ versus N-N, A-A$_{same}$ versus A-N, A-A$_{other}$ versus N-N, A-A$_{other}$ versus A-N, N-N versus A-N, respectively. Wilcoxon rank sums test with Bonferroni post hoc correction). Data are presented as mean ± SEM from 11 imaging sessions on 7 animals. Data from astrocytic recording comprises 36 cells in which there was significant spatial modulation in at least 1 ROI. The data presented in this figure can be found in S4 Data. ROI, region of interest; SEM, standard error of the mean.
(TIFF)

**S15 Fig. Pairwise correlation of calcium signals and difference in field position as a function of pairwise distance. (A)** The distance (d) between the centers 2 ROIs comprising a pair is computed for all astrocytic (top) and neuronal (bottom) ROIs. **(B, C)** Pearson correlation (B) and difference between response field position (C) as a function of pairwise distance for pairs of astrocytic ROIs with reliable spatial information (cyan) and pairs of neuronal ROIs with reliable spatial information (purple). Data are expressed as mean ± SEM from 11 imaging sessions on 7 animals. (A) $p$ = 8E-4, $p$ = 8E-4, $p$ = 1E-4, $p$ = 1E-3, $p$ = 1E-3, $p$ = 1E-3, $p$ = 8E-4, and $p$ = 2E-1 for 10, 30, 70, 90, 110, 130, and 150 μm pairwise distances, respectively. Two-sample Kolmogorov–Smirnov test with Bonferroni post hoc correction. (B) $p$ = 1, $p$ = 1, $p$ = 0.7, $p$ = 1, $p$ = 1, $p$ = 1, $p$ = 0.2, and $p$ = 0.2 for 10, 30, 70, 90, 110, 130, and 150 μm pairwise distances, respectively. Two-sample Kolmogorov–Smirnov test with Bonferroni post hoc correction. Data from astrocytic recording comprises 36 cells in which there was significant spatial modulation in at least 1 ROI. The data presented in this figure can be found in S5 Data. ROI, region of interest; SEM, standard error of the mean.
(TIFF)

**S16 Fig. Precision and stability of neuronal and astrocytic spatial responses. (A)** Spatial precision index for simultaneously recorded neuronal and astrocytic response fields (mean ± SEM; neuronal responses 7.5E-2 ± 1.6E-2; astrocytic responses 4.1E-2 ± 0.2E-2; $p$ = 4.6E-2 Wilcoxon signed rank test; data from 11 imaging sessions on 7 animals). **(B)** Fraction of neuronal and astrocytic ROIs showing reliable spatial information and stable response field (mean ± SD; neurons 0.16 ± 0.09; astrocytic responses 0.08 ± 0.07; $p$ = 2.9E-1 Wilcoxon signed rank test; data from 11 imaging sessions on 7 animals). The data presented in this figure can be found in S5 Data. ROI, region of interest; SD, standard deviation; SEM, standard error of the mean.
(TIFF)

**S17 Fig. The majority of spatial information in astrocytes and neurons is genuine spatial information that cannot be explained by tuning to visual cues. (A, B)** Fraction of astrocytic (A) and neuronal (B) ROIs encoding reliable spatial information showing a significant decrease in their information content when position is shuffled within visual cues. Shuffling position within individual visual cues decoupled spatial information encoded in the astrocytic response from the information related to visual cues identity (see Methods). The fraction of ROIs showing significant information loss is shown as function of the number of position bins used to compute mutual information. Data are presented as mean ± SEM from 11 experimental sessions in 7 animals, ($N$ = 76 for astrocytic ROIs, $N$ = 335 for neuronal ROIs, binomial test, see S8 Table). The data presented in this figure can be found in S5 Data. ROI, region of

interest; SEM, standard error of the mean.
(TIFF)

**S18 Fig. Visual cues identity does not explain animal's position decoding from neither astrocytic nor neuronal calcium signals. (A)** Confusion matrices of an SVM classifier decoding the mouse's position using population vectors comprising either astrocytic (top) or neuronal (bottom) ROIs in which position was shuffled within visual cues. Shuffling position within visual cues decoupled spatial information encoded in the population vector from the information related to visual cues identity (see Methods). The true position of the animal is shown on the x-axis and the decoded position on the y-axis. Gray scale indicates the percentage of occurrence of each matrix element. Results are shown for various spatial granularities (G = 9, 12, 15, 18). **(B)** Decoded information from astrocytic population vectors as a function of decoding granularity on real data (white) and for data in which position is shuffled within visual cues (gray, see Methods). **(C)** Fraction of genuine spatial information in astrocytic population vectors computed shuffling position within individual visual cues. Results are shown as a function of decoding granularity. **(D, E)** Same as in (B, C) but from population vectors comprising neuronal ROIs. In all panels, data are shown as mean ± SEM and were obtained from 11 imaging sessions in 7 animals (see also S9 Table). The data presented in this figure can be found in S5 Data. ROI, region of interest; SEM, standard error of the mean; SVM, support vector machine. (TIFF)

**S19 Fig. Astrocytes and neurons encode complementary and synergistic spatial information. (A)** Mutual information about position encoded by pairs of ROIs (I) is shown in comparison to the sum ($I_{LIN}$) and to the maximum ($I_{MAX}$) of the information separately encoded by each component of the pair. A-A, pair composed of 2 astrocytic ROIs; N-N, pair composed of 2 neuronal ROIs; A-N, mixed pair composed of one astrocytic and one neuronal ROI. For this analysis, the values of information were computed using the "shuffled" bias correction procedure (see methods) which overcorrects the bias inducing an underestimation of I (I versus $I_{LIN}$: A-A: $p = 1E-2$, N-N: $p = 7E-3$, A-N: $p = 1E-3$; I versus $I_{MAX}$: A-A: $p = 5E-3$, N-N: $p = 1E-3$, A-N: $p = 1E-3$, Wilcoxon signed rank test, see also S3 Table). Data are represented as mean ± SEM from 11 imaging sessions in 7 animals. The data presented in this figure can be found in S5 Data. ROI, region of interest, SEM, standard error of the mean. (TIFF)

**S20 Fig. Position-dependent correlations contribute to synergistic information encoding. (A, B)** Information breakdown for the different types of ROI pairs: 2 astrocytic ROIs (A-A), 2 neuronal ROIs (N-N), or one astrocytic and one neuronal ROI (A-N). Pairs were classified as synergistic (B) based on the value of ΔI (see Methods). I (white) is the mutual information about position encoded by the pair. $I_{LIN}$ (gray) is the sum of the mutual information about position independently encoded in the response of each member of the pair. $I_{SS}$ (red) is the redundant information component quantifying similarity in the responses of the members of the pair. $I_{CI}$ (green) and $I_{CD}$ (blue) quantify the information contribution of correlation independent or dependent on position, respectively. Data are represented as mean ± SEM and were collected in 11 imaging sessions on 7 animals. The data presented in this figure can be found in S5 Data. ROI, region of interest; SEM, standard error of the mean. (TIFF)

**S21 Fig. Correlation between astrocytes and neurons is animal's position-dependent. (A–D)** Scatterplot of the absolute value of Pearson correlation outside the response field against the absolute value of Pearson correlation inside the response field for pairs comprising one astrocytic and one neuronal ROI. Black open dots show averages of each imaging session, the

red cross shows the mean ± SEM (A, B) Correlations were measured for all possible pairs. In (A), correlations are computed with respect to astrocytic response field (mean correlation inside the response field $0.11 \pm 0.01$; mean correlation outside the response field $0.07 \pm 0.01$, $p = 6.4E\text{-}3$ Wilcoxon rank sums test). In (B), correlations are computed with respect to neuronal response field (mean correlation inside the response field $0.12 \pm 0.01$; mean correlation outside the response field $0.07 \pm 0.01$, $p = 1.1E\text{-}3$ Wilcoxon rank sums test). (C, D) Same as (A, B) but correlations were computed only on synergistic pairs based on the value of ΔI (see Methods, Fig 6, and S11 Fig). In (C), correlations are computed with respect to astrocytic response field (mean correlation inside the response field $0.12 \pm 0.01$; mean correlation outside the response field $0.09 \pm 0.01$, $p = 7.8E\text{-}3$ Wilcoxon rank sums test). In (D), correlations are computed with respect to neuronal response field (mean correlation inside the response field $0.13 \pm 0.01$; mean correlation outside the response field $0.08 \pm 0.01$, $p = 1.8E\text{-}3$ Wilcoxon rank sums test). For each pair of ROIs, correlations were computed averaging 100 resampling to compensate unbalanced observations inside and outside the response field. Data from 11 imaging sessions on 7 animals. The data presented in this figure can be found in S5 Data. ROI, region of interest; SEM, standard error of the mean.
(TIFF)

**S22 Fig. Decoding the animal's position from neuronal and astrocytic population vectors.** **(A)** Confusion matrices of an SVM classifier decoding the mouse's position using population vectors comprising neuronal (top), astrocytic (middle), and neuronal + astrocytic ROIs (bottom) for various spatial granularities (G = 4, 8, 12, 16, 20, 24). The true position of the animal is shown on the x-axis and the decoded position on the y-axis. Gray scale indicates the percentage of occurrence of each matrix element. **(B)** Decoded mutual information between predicted and real position in the linear track and **(C)** decoding accuracy for the different population vectors as a function of spatial granularity. In B and C, asterisks indicate significance against chance level (S5 and S10 Tables). Data are displayed as mean ± SEM and were collected in 11 imaging sessions from 7 animals. The data presented in this figure can be found in S5 Data. ROI, region of interest; SEM, standard error of the mean; SVM, support vector machine.
(TIFF)

**S1 Table. Outline and summary of experiments.**
(DOCX)

**S2 Table. Hypothesis testing: decoding performance about animal's spatial location from astrocytic calcium signals during monodirectional virtual navigation.** *p*-values for one-tailed nonparametric permutation tests as a function of decoding granularity for decoded information (see Fig 3B) and decoding accuracy (S6 Fig). For each imaging session and each granularity, null distributions were obtained with 1,000 and 500 iterations to estimate chance level and trial shuffling, respectively (see Methods). Data from 7 imaging sessions from 3 animals. The data presented in this figure can be found in S1 Data.
(DOCX)

**S3 Table. Complementary and synergistic spatial information encoding in astrocytic and neuronal calcium signals.** Information about position carried by pairs of ROIs (I) is compared to the sum ($I_{LIN}$) or to the maximum ($I_{MAX}$) of the information separately encoded by each member of the pair. A-A, pair composed of two astrocytic ROIs; N-N, pair composed of two neuronal ROIs; A-N, mixed pair composed of one astrocytic and one neuronal ROI. We summarize mean difference between groups, confidence interval limits, Cohen d effect size estimate, and *p*-value for Wilcoxon signed rank test. Information measures were corrected using two bias correction procedures, QE, and shuffled. Data are from 11 imaging sessions on 7

animals. The data for this table can be found in S1 Data and S5 Data. QE, quadratic extrapolation; ROI, region of interest.
(DOCX)

**S4 Table. Comparison of decoding information about animal's spatial location from neuronal and astrocytic population vectors.** *p*-values for two-tailed paired *t* tests with Bonferroni correction for decoded information of animal's spatial location from population vectors comprising all astrocytic ROIs versus all ROIs of both types (top row) and all neuronal ROIs versus all ROIs of both types (bottom row) during monodirectional virtual navigation shown in Fig 6. Data from 11 imaging sessions from 7 animals. The data for this table can be found in S1 Data and S5 Data. ROI, region of interest.
(DOCX)

**S5 Table. Hypothesis testing: decoding information about animal's spatial location from neuronal and astrocytic population vectors.** *p*-values for one-tailed nonparametric permutation tests for decoding information from population vectors comprising either all astrocytic (top row), all neuronal (middle row), or ROIs of both types (bottom row) during monodirectional virtual navigation (see Fig 6 and S13 Fig). Significance levels are reported as a function of decoding granularity. For each imaging session and each granularity, null distributions were obtained with 1,000 and 500 iterations to estimate chance level and trial shuffling, respectively (Methods). Data from 11 imaging sessions from 7 animals. The data for this table can be found in S1 Data and S5 Data. ROI, region of interest.
(DOCX)

**S6 Table. Hypothesis testing: decoding performances about animal's spatial location from astrocytic calcium signals during bidirectional virtual navigation.** *p*-values for one-tailed nonparametric permutation tests as a function of decoding granularity for decoded information (see S7B and S7F Fig) and decoding accuracy (see S7C and S7G Fig). Decoding performance is reported for forward- and backward-running directions (see S7 Fig). For each imaging session and each granularity, null distributions were obtained with 1,000 iterations to estimate chance level (Methods). Data from 15 imaging sessions in 4 animals for forward-running direction. Data from 17 imaging sessions in 4 animals for backward-running direction. The data for this table can be found in S3 Data.
(DOCX)

**S7 Table. Pairwise correlations of calcium signals during virtual navigation.** Descriptive statistics and confidence intervals estimation for pairwise Pearson correlation. Mean, SEM, 95% confidence interval limits, and *p*-value for Wilcoxon rank sums test for $H_0 = 0$ are shown. Pairs were composed either of 2 astrocytic ROIs belonging to the same astrocyte (A-A$_{same}$), 2 astrocytic ROIs belonging to the different astrocytes (A-A$_{other}$), 2 neuronal ROIs (N-N), or one astrocytic and one neuronal ROI (A-N). Correlation was measured for ROI pairs with reliable spatial information or for all possible pairs. Data are from 11 imaging sessions on 7 animals. The data for this table can be found in S4 Data. ROI, region of interest; SEM, standard error of the mean.
(DOCX)

**S8 Table. Hypothesis testing: visual cue identity does not explain neither astrocytic nor neuronal spatial tuning during virtual navigation.** *p*-values for binomial tests for astrocytic (top row) or neuronal (bottom row) ROIs encoding reliable spatial information showing a significant decrease in their information content when position was shuffled within individual visual cues (see also S14 Fig). Significance levels are reported as a function of the number of

position bins ($N_S$). For each imaging session and each $N_S$, $I_V$ distributions were obtained with 100 iterations in which position was shuffled within visual cues to estimate average $I_V$ (see also Methods). Data from 11 imaging sessions from 7 animals. The data for this table can be found in S5 Data. ROI, region of interest.
(DOCX)

**S9 Table. Hypothesis testing: decoding genuine spatial information from astrocytic and neuronal calcium signals.** *p*-values for Wilcoxon signed rank tests for decoding information from population vectors comprising either all astrocytic (top row) or all neuronal (bottom row) ROIs during monodirectional virtual navigation (see S21 Fig). Significance levels are reported as a function of decoding granularity. For each imaging session and each granularity, $I_V$ distributions were obtained with 500 iterations in which position was shuffled within visual cues to estimate average $I_V$ (see Methods). Data from 11 imaging sessions from 7 animals. The data for this table can be found in S5 Data. ROI, region of interest.
(DOCX)

**S10 Table. Hypothesis testing: spatial decoding accuracy from neuronal and astrocytic population vectors.** *p*-values for one-tailed nonparametric permutation tests for decoding accuracy from population vectors comprising either all astrocytic ROIs (top row), all neuronal ROIs (middle row), or all ROIs of both types (bottom row) during monodirectional virtual navigation (see S13 Fig). Significance levels are reported as a function of decoding granularity. For each imaging session and each granularity, null distributions were obtained with 1,000 iterations to estimate chance level (Methods). Data from 11 imaging sessions from 7 animals. The data for this table can be found in S1 Data and S5 Data. ROI, region of interest.
(DOCX)

**S1 Data. Source data for Figs 1–6.**
(ZIP)

**S2 Data. Source data for S1–S8 Figs.**
(ZIP)

**S3 Data. Source data for S9–S13 Figs.**
(ZIP)

**S4 Data. Source data for S14 Fig.**
(ZIP)

**S5 Data. Source data for S15–S22 Figs.**
(ZIP)

## Acknowledgments

We thank C. Harvey, G. Edgerton, and B. Mensh for helpful comments on the manuscript; L. Sità and F. Succol for technical help; O. Yizhar for sharing the jRCaMP1a construct; A. Attardo for help with surgical procedures; and A. Contestabile for help with the virus production.

### Extended data

This work has extended data in S1–S22 Figs and S1–S10 Tables.

## Author Contributions

**Conceptualization:** Stefano Panzeri, Tommaso Fellin.

**Data curation:** Sebastiano Curreli, Jacopo Bonato, Sara Romanzi.

**Formal analysis:** Sebastiano Curreli, Jacopo Bonato, Sara Romanzi.

**Funding acquisition:** Stefano Panzeri, Tommaso Fellin.

**Investigation:** Sebastiano Curreli, Jacopo Bonato, Sara Romanzi.

**Methodology:** Sebastiano Curreli, Jacopo Bonato, Stefano Panzeri, Tommaso Fellin.

**Project administration:** Stefano Panzeri, Tommaso Fellin.

**Resources:** Stefano Panzeri, Tommaso Fellin.

**Software:** Sebastiano Curreli, Jacopo Bonato.

**Supervision:** Stefano Panzeri, Tommaso Fellin.

**Validation:** Sebastiano Curreli, Jacopo Bonato, Sara Romanzi.

**Visualization:** Sebastiano Curreli, Jacopo Bonato, Sara Romanzi.

**Writing – original draft:** Sebastiano Curreli, Jacopo Bonato, Sara Romanzi, Stefano Panzeri, Tommaso Fellin.

**Writing – review & editing:** Sebastiano Curreli, Jacopo Bonato, Stefano Panzeri, Tommaso Fellin.

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
