## [Editor Report · Decision Letter 0]

18 Jun 2021

Dear Dr Fellin, 

Thank you for submitting your manuscript entitled "Glial place cells: complementary encoding of spatial information in hippocampal astrocytes" for consideration as a Research Article by PLOS Biology.

Your manuscript has now been evaluated by the PLOS Biology editorial staff as well as by an academic editor with relevant expertise and I am writing to let you know that we would like to send your submission out for external peer review.

Please re-submit your manuscript within two working days, i.e. by Jun 22 2021 11:59PM.

Kind regards,

Lucas Smith

Associate Editor

PLOS Biology

lsmith@plos.org

---

## [Decision Letter · Decision Letter 1]

19 Jul 2021

Dear Dr Fellin,

Thank you very much for submitting your manuscript "Glial place cells: complementary encoding of spatial information in hippocampal astrocytes" for consideration as a Research Article at PLOS Biology. Your manuscript has been evaluated by the PLOS Biology editors, an Academic Editor with relevant expertise, and by several independent reviewers.

The reviews of your manuscript are appended below. As you will see, while the reviewers find the study interesting, they have also raised a number of concerns that will need to be addressed before we can consider your manuscript for publication. In particular, the reviewers have commented that some of the conclusions made in the study are not adequately supported by the data, and editorially we find the comments from Reviewer 3 to be most concerning. In light of the reviews, we will not be able to accept the current version of the manuscript, but we would welcome re-submission of a much-revised version that takes into account the reviewers' comments and addresses their specific concerns. We cannot make any decision about publication until we have seen the revised manuscript and your response to the reviewers' comments. Your revised manuscript is also likely to be sent for further evaluation by the reviewers.

We expect to receive your revised manuscript within 3 months. 

**IMPORTANT - SUBMITTING YOUR REVISION**

*Re-submission Checklist*

*Published Peer Review*

*PLOS Data Policy*

*Blot and Gel Data Policy*

Sincerely,

Lucas Smith

Associate Editor

PLOS Biology

lsmith@plos.org

REVIEWS:

Reviewer #1: In this manuscript, authors have investigated the calcium activity of hippocampal astrocytes and neurons relate to the spatial information of mice navigating a virtual. To investigate this, authors have used genetically-encoded calcium indicators and simultaneous two-photon calcium imaging of hippocampal astrocytes and neurons in awake mice.

Authors developed and used a meticulous and complex analysis of the calcium activity of astrocyte somas and processes as well as neurons in relation to the virtual spatial location of the animal. The study provides an exhaustive characterization of the signals, which can be summarized in two important messages: 1) astrocyte activity in the somas and processes is modulated by the spatial position of the mice, indicating that they encode spatial information; 2) astrocytes have broader response field width and a different distribution of field position that neurons, suggesting a complementary and synergistic spatial information processed by astrocytes and neurons.

This is an exciting and elegant study that adds valuable information regarding a relevant topic in current neuroscience, the physiology of astrocytes and their active involvement in brain function. More specifically, the present demonstration that astrocytes encode spatial information, i.e., astrocytes act like place cells, is a break-through finding that change our view of information processing in the brain.

The manuscript presents a very exhaustive and deep study that presents novel and interesting results, which are very convincing. Furthermore, the study is methodologically and technically adequate, with elegant and appropriate experimental design to provide robust results. The conclusions reached are well-supported by the experimental data.

Therefore, I have few comments about present manuscript. While it presents many strengths, I have also found some weaknesses that need to be addressed to improve the already high quality of the manuscript. 

In summary, the results presented are novel and relevant. I believe that present manuscript will positively and strongly impact the neuroscience field. It has high merit to be published in PLOS Biology. 

Specific comments

1. When analyzing calcium events, authors describe in Methods: "for each trace, the standard deviation (σ1) of the signal was computed and values crossing threshold set at ± σ1 were removed from the trace. This procedure excluded large transients". Authors should provide rationale about why these large events were excluded. If they are not excluded, would the results presented still be valid?

2. To simultaneously monitor astrocyte and neuronal activity, authors delivered virus containing different genetically encoded calcium indicators under cell-specific promoters that are expected to provide cell-specific expression. Although this approach is widely and confidently used, authors should provide immunohistochemistry quantification that this is the case in their own hands. This is a very simple control that is needed to support the validity of the data.

3. The analysis used is certainly complex and represents a tour de force, probably necessary to reveal the results obtained. It is perfectly defined and provide robust results. Yet, one could wonder whether such results may be an "artifact" of the analysis. Authors could use some controls to discard this possibility. For example, how is the astrocyte and neuronal activities in a static position with absence of virtual navigation? This is probably an unnecessary control, but it may enhance the robustness of the results by discarding some potential analytical artifacts.

4. In several instances, authors seem to imply that astrocytic encoding of spatial information is involved in the brain information processing. For example, in page 4, line 78: "Thus, astrocytic calcium signaling participates in encoding spatial information in the hippocampus". While I am convinced by the results that astrocytes encode spatial information, the fact that this is translated to the information processing of the hippocampus has not been actually tested. This is an interesting hypothesis, but it would require additional studies, certainly out of the scope of the present manuscript, such as the specific manipulation of the astrocyte activity and analysis of the outcome. Unless authors perform such studies, they should town down the interpretation of the results in the Results section, restricting such hypothetical ideas to the Discussion.

Minor comments

1. The wording of the title may be confusing. "Glial place cells" is not completely proper. There are several types of glial cells, but the study is focused on astrocytes. 

Reviewer #2: This paper shows for the first time that astrocytes in the CA1 region of the hippocampus have place fields that tile a virtual environment in similar ways to neuronal place cells. Further, they show that astrocytic processes from the same astrocytes can have differences in their place field locations relative to the cell body. Lastly, they claim that astrocytes carry unique spatial information from neuronal populations that, synergistically with neurons, enhances the overall spatial encoding properties of the CA1 region. This is new information that makes a significant contribution to the field and opens new avenues to explore as astrocytes have been left out of most theories of hippocampal function as we did not know their encoding properties. However, some of their conclusions are supported by data that I believe needs further analysis to be convincing. 

These comments are in the order in which I read the manuscript, not in order of significance:

* They show mean astrocyte activity as a function of position (Fig, 1E), but it would be informative to see trial-to-trial traces to get an idea of their variability considering this is the first time the field is seeing astrocytic place fields (PFs).

* Along these lines, they should quantify trial-to-trial parameters of these astrocyte PFs such as overdispersion1 and precision2 and compare them to neuronal PFs. 

* And what about PF stability? Do they change, shift, or remap throughout a session? Neuronal PFs have been shown to shift in CA1 and even remap over longer time periods. It would be good to analyze astrocytic PF dynamics in this way. They could be more or less stable than neuronal PFs, and that would be important to know.

* A recent paper on bioRxiv3 shows CA1 astrocytes have a ramping to reward signal in an almost identical task as shown here. Do the authors also see these signals? In addition, is there an over-representation of the reward zone in astrocytic PFs similar to what is seen with neuronal PFs? 

* What is their interpretation of what underlies astrocytic calcium signals? What is the source of calcium? What membrane potentials are likely leading to the calcium signals they see? What is thought to drive depolarizations in astrocytes and how do they differe to the inputs driving CA1 pyramidal cells? A discussion on how the authors think about these signals is needed.

* I think it is hard to interpret comparisons between calcium dynamics in Astrocytes and neurons (Fig. 4) as one is in due to underlying action potentials and the other is due to graded changes in Vm. Calcium buffering is also likely very different in astrocytes versus neurons (which they do briefly mention in the discussion). Also, GCaMP (astrocytes) and RGeCO (neurons) have different kinetics. Therefore, concluding that PF widths or any other properties are different is difficult and needs further experimental evidence (ephys). I am not asking them to do these experiments as they are extremely difficult, but I am asking them to temper their conclusions based on these confounds or do more analysis to better support their claims here.

* Supp Fig 9 shows the mean correlation of A-N pairs of cells with high spatial info, with the conclusion that they are significantly correlated above 0 (line 144). However, the effect size is very small. It would be much better to quantify this effect size using different analysis. I recommend the authors conduct estimation approaches on these datasets4. This emphasizes reporting effect sizes with expressions of uncertainty (interval estimates) to make better inferences: it pushes back against over-confident claims from inadequate samples, improves comparisons of results across contexts, and normalizes the publication of negligible effects4.

* Lines 145-146 states "Correlation among pairs of astrocytic ROIs was generally higher than correlation among pairs of neuronal ROIs". But isn't this to be expected given the population of astrocytic ROIs comprises processes that are part of the same astrocyte? And even though the authors have found processes from the same astrocyte can have different PFs, they are still likely to be more correlated that PFs from other astrocytes. It is therefore not a fair comparison between these neuronal and astrocyte populations. 

* Fig 5A, again, the effect sizes for comparisons are very small in some cases. Basing conclusions on these is problematic. Estimation approaches should again be used here. This may influence a major conclusion of the paper which is stated in Line 160: "information carried by the pairs was also synergistic". With such small effect sizes the authors need to be very careful about making such a conclusion. Doing the estimation analysis will help in this regard and show how big or small these differences are, and conclusions should be based on that.

* Lines 178-180 "Complementary and synergistic spatial information encoding in mixed pairs suggests that the astrocytic network carries additional information unique from that encoded in neuronal circuits also at the whole population level." But is this necessarily true? Does it really mean the astrocytic information is "unique" or is the information the same, it's just adding more of it to help the decoder? The authors should explain their thinking a little more on this point. Astrocytes are encoding place, so in that sense they are not unique. And decoder performance is always a function of the number of cells used. For instance, if they recorded from more place cells their decoder would perform better. Lines 187-188 states "This result proved that the population of astrocytic ROIs carries information not found in neurons or their interactions." But the authors didn't record from all neurons. And again, if they added more neurons their decoder would perform better. Would they then conclude that the extra neurons they recorded from were unique?

* Lines 209-211 "response field position was differentially distributed in astrocytes compared to neurons, suggesting that CA1 astrocytes do not merely mirror position information encoded in CA1 pyramidal neurons." I think this is problematic because the astrocyte population is comprised of processes from the same astrocytes, whereas the neurons are independent somas. Furthermore, the population of independent astrocytes is much smaller than the population of neurons (see fig 2B versus 4C, right). Even though the authors have shown that processes from the same astrocyte can have distinct PFs, they are still more correlated than independent astrocytes. These factors will influence the population map of the corridor, making it difficult to conclude that astrocytes are differentially distributed. Again, they also did not record from all neurons or all astrocytes, and the numbers of independent astrocytes and neurons is not matched.

* I think the authors should expand their discussion to include what they think the downstream effects of the astrocytic spatial code is for? Obviously, the neuronal spatial code is projected out to other brain regions and used there for certain computations. But the astrocytic spatial code remains within the hippocampus. What is its ultimate function? Is the brain using this information to actually help encode external space, and if so, how? Or possibly it is serving modulatory role within the CA1 itself. If we were to inactivate the astrocytes, what would be the prediction? Would the neuronal spatial code be affected? An expansion on these ideas would be very useful for the discussion section.

1 Fenton, A. A. et al. Attention-like modulation of hippocampus place cell discharge. J Neurosci 30, 4613-4625, doi:10.1523/JNEUROSCI.5576-09.2010 (2010).

2 Sheffield, M. E. & Dombeck, D. A. Calcium transient prevalence across the dendritic arbour predicts place field properties. Nature 517, 200-204, doi:10.1038/nature13871 (2015).

3 Doron, A. et al. Hippocampal Astrocytes Encode Reward Location. bioRxiv, 2021.2007.2007.451434, doi:10.1101/2021.07.07.451434 (2021).

4 Calin-Jageman, R. J. & Cumming, G. Estimation for Better Inference in Neuroscience. eNeuro 6, doi:10.1523/ENEURO.0205-19.2019 (2019).

Reviewer #3: Curreli et al. investigated how hippocampal astrocyte calcium signals are involved in spatial information encoding. The authors expressed a genetic calcium indicator in the hippocampal astrocytes of adult mice and found that astrocyte calcium activities from somata and processes both respond to spatial cues in a virtual reality environment with two-photon in vivo imaging. The authors further examined the neuronal activities in the same virtual reality setting and showed that calcium signals of astrocytes and neurons appear correlated and convey synergistic information. 

Increasing evidence has demonstrated the importance of astrocyte calcium signaling in regulating neural circuits and animal behavior. Yet, whether astrocytes utilize their diverse calcium signals to encode spatial information is still unclear. Thus, the work by Curreli et al. is novel from this perspective. However, some of the conclusions drawn from the data are open to interpretation and there are several major issues that the authors need to adequately address. The authors also did not rule out the possibility that astrocytes solely imitate or react to changes of neighboring neurons and therefore seem to be responsive to spatial cues. Without further supporting evidence, the discovery of "astrocytic place cells" is unconvincing.

Major points:

1. The authors imaged astrocyte calcium signals in a virtual reality with a monodirectional or bidirectional corridor, which significantly limits the exploration dimension and spatial information that are normally contained in a real environment. Therefore, instead of being responsive to the spatial information as suggested by the authors, detected astrocyte calcium signals could be induced simply by the visual stimuli such as images, lighting, shapes etc from the screen of the virtual reality setting. Without further evidence, it is unconvincing to conclude that there are "astrocytic place cells" and their calcium signals encode spatial information. 

2. Astrocytes display spontaneous calcium signals that occur in the absence of external stimuli. To rule out the possibility that observed astrocyte calcium responses in the virtual reality are not due to random activities, additional controls are warranted. For example, what are the baseline calcium activities of the previously identified "place cells" without virtual reality? How do "astrocytic place cells" respond to the same virtual reality pattern, e.g. record with only displaying the vertical lines? How do they react to alternating virtual reality patterns, e.g. record with mesh pattern -> vertical lines -> mesh pattern? These experiments will help to clarify whether the calcium responses of astrocytes are specific to certain spatial cues, which is expected for a place cell.

3. In Fig. 4, both neurons and astrocytes were imaged "in the CA1 pyramidal layer" and their calcium signals were analyzed to reveal the spatial correlation. However, there are very few astrocytes present in the pyramidal layer, so it is unclear how both cell types were imaged simultaneously with two-photon microscopy. If neurons and astrocytes were indeed captured from the same focal plane, then the authors need to provide representative images. Fig. 4A only shows astrocytes but not neurons. If, however, neurons and astrocytes were sampled from different focal planes, then the authors have to indicate so and take the depth difference into consideration for subsequent spatial analysis in Fig. 4F. The current presentation and description are misleading.

Minor points:

1. The cortex above the hippocampus was aspirated to facilitate in vivo imaging. However, astrocytes may become reactive and change calcium signals due to the invasive surgical procedure, this possibility should be tested with astrocyte reactive marker staining or morphological analysis. 

2. As the authors already pointed out, the properties of astrocyte calcium signals from different cellular compartments (e.g. somata vs. branches vs. processes) are fundamentally different. It is interesting that the distribution of field position of ROIs from soma are similar to the ones from the processes. However, it is unclear how many ROIs were identified from the processes belonging to the same astrocytes. It is not surprising that ROIs from the same cellular compartment of the same astrocytes are highly correlated, as shown in Fig. 2F and Fig. 2G. Thus, the authors should separate intracellular correlation from intercellular correlation. Combing all ROIs together may overlook the diversity of astrocyte calcium signals. In addition to the number of ROIs and mice, the authors should also clarify the number of cells in the analysis. 

3. In Fig 3B, because the "chance level" is extremely low so it doesn't appear in the bar graph. The y axis should be adjusted to show all the data, e.g. starting from -0.1.

---

## [Decision Letter · Decision Letter 2]

13 Dec 2021

Dear Dr Fellin,

Thank you for submitting your revised Research Article entitled "Glial place cells: complementary encoding of spatial information in hippocampal astrocytes" for publication in PLOS Biology. I have now obtained advice from the original reviewers and have discussed their comments with the Academic Editor.  

The reviews are appended below and as you will see, the reviewers are largely satisfied by the revision. However Reviewer 3 has raised an important lingering concern which needs to be addressed. Based on the reviews, we will probably accept this manuscript for publication, provided you satisfactorily address the remaining point raised by the Reviewer 3 with a revision that we think should not take very long. **IMPORTANT Please also make sure to address the following data and other policy-related requests:

1) ETHICS REQUEST: In the methods section of your manuscript, please include the approval number for the animal care and use protocol approved by the National Council on Animal Care of the National Council on Animal Care of the Italian Ministry of Health.

2) DATA REQUEST: in the materials and methods section of your manuscript, I noticed that you discuss data not presented in the manuscript (data not shown; page 39). Please note that per journal policy, we do not allow the mention of "data not shown", "personal communication", "manuscript in preparation" or other references to data that is not publicly available or contained within this manuscript. Please either remove mention of these data or provide figures presenting the results and the data underlying the figure(s).

3) DATA REQUEST: Thank you for providing the underlying data for each figure as supplementary excel files. Please ensure that the figure legends in your manuscript include information on where the underlying data can be found. For example, to each figure legend you can say "the data underlying this figure can be found in ___" (referencing the relevant file). 

4) TITLE: After some discussion within the team, we think the title of your manuscript should be edited slightly to be more streamlined and avoid punctuation. If you agree, we would suggest changing it to something like "Hippocampal astrocytes act as glial place cells that encode spatial information to complement surrounding neurons"

We expect to receive your revised manuscript within three weeks, although do let us know if you would like an extension. 

*Published Peer Review History*

*Early Version*

Sincerely,

Lucas Smith, Ph.D.,

Associate Editor,

lsmith@plos.org,

PLOS Biology

Reviewer Comments: 

Reviewer #1: Authors have adequately addressed the concerns expressed about the previous version. 

The new data and the clarifications provided further and strongly support the important conclusions reached.

Therefore, I have no further comments and I confirm my initial assessment of the previous version of the manuscript:

This is an exciting and elegant study that adds valuable information regarding a relevant topic in current neuroscience, the physiology of astrocytes and their active involvement in brain function. More specifically, the present demonstration that astrocytes encode spatial information, i.e., astrocytes act like place cells, is a break-through finding that change our view of information processing in the brain.

The manuscript presents a very exhaustive and deep study that presents novel and interesting results, which are very convincing. Furthermore, the study is methodologically and technically adequate, with elegant and appropriate experimental design to provide robust results. The conclusions reached are well-supported by the experimental data.

In summary, the results presented are novel and relevant. I believe that present manuscript will positively and strongly impact the neuroscience field. It has high merit to be published in PLOS Biology. 

Reviewer #2: The authors addressed all my concerns. I have no further comments.

Reviewer #3: Curreli et al. has included additional analyses in the revised manuscript to address most of my questions. One issue that I encourage the authors to seriously consider, which was also raised by other reviewers is to tune down their claims that astrocytes and neurons encode or process different spatial information. Since astrocytes are fundamentally different from neurons, the fact that astrocyte calcium signaling exhibit distinct properties from neurons could likely be due to intrinsic molecular machinery operated in the cells. That is, astrocyte and neuronal calcium signaling can encode the same information, but display in a distinct manner. In fact, transcriptomic and proteomic studies have all supported this notion. For instance, the authors concluded based on Fig. 6 that "This result proved that the population of astrocytic ROIs carries information not found in neurons or their interactions." In order to experimentally "prove" this statement, the authors should manipulate calcium signaling in either astrocytes or neurons, and analyze the consequences of animal's navigational abilities, which I believe is beyond the scope of this study. Thus, the authors should rephrase their statements to avoid misleading the readers.

---

## [Editor Report · Decision Letter 3]

5 Jan 2022

Dear Dr Fellin,

Happy New Years, and thank you again for your patience while we were off on our winter break. I am writing because we have now had a chance to evaluate your revision. On behalf of my colleagues and the Academic Editor, Thomas Klausberger, I am pleased to say that we can in principle accept your Research Article "Complementary encoding of spatial information in hippocampal astrocytes" for publication in PLOS Biology.

Please note, that before we can schedule your manuscript for publication we will need you to address any remaining formatting and reporting issues. These will be detailed in an email that will follow this letter and that you will usually receive within 2-3 business days, during which time no action is required from you.

PRESS

Sincerely, 

Lucas Smith, Ph.D. 

Senior Editor 

PLOS Biology

lsmith@plos.org